# Neutrophils homing into the retina trigger pathology in early age-related macular degeneration

Sayan Ghosh[1], Archana Padmanabhan[2], Tanuja Vaidya[2], Alan M. Watson[3], Imran A. Bhutto[1], Stacey Hose[1], Peng Shang[1], Nadezda Stepicheva[1], Meysam Yazdankhah[1], Joseph Weiss[1], Manjula Das[4], Santosh Gopikrishna[2], Aishwarya[2], Naresh Yadav[2], Thorsten Berger[5], Tak W. Mak[5], Shuli Xia[6], Jiang Qian[7], Gerard A. Lutty[7], Ashwath Jayagopal[8,9], J. Samuel Zigler Jr[7], Swaminathan Sethu[2], James T. Handa[7], Simon C. Watkins (ID)[3], Arkasubhra Ghosh (ID)[2] & Debasish Sinha[1,7]

Age-related macular degeneration (AMD) is an expanding problem as longevity increases worldwide. While inflammation clearly contributes to vision loss in AMD, the mechanism remains controversial. Here we show that neutrophils are important in this inflammatory process. In the retinas of both early AMD patients and in a mouse model with an early AMD-like phenotype, we show neutrophil infiltration. Such infiltration was confirmed experimentally using ribbon-scanning confocal microscopy (RSCM) and IFNλ− activated dye labeled normal neutrophils. With neutrophils lacking lipocalin-2 (LCN-2), infiltration was greatly reduced. Further, increased levels of IFNλ in early AMD trigger neutrophil activation and LCN-2 upregulation. LCN-2 promotes inflammation by modulating integrin β1 levels to stimulate adhesion and transmigration of activated neutrophils into the retina. We show that in the mouse model, inhibiting AKT2 neutralizes IFNλ inflammatory signals, reduces LCN-2-mediated neutrophil infiltration, and reverses early AMD-like phenotype changes. Thus, AKT2 inhibitors may have therapeutic potential in early, dry AMD.

[1] Department of Ophthalmology, University of Pittsburgh School of Medicine, Pittsburgh, PA, USA. [2] Narayana Nethralaya Foundation, Bengaluru, India. [3] Center for Biologic Imaging and Department of Cellular Biology, University of Pittsburgh School of Medicine, Pittsburgh, PA, USA. [4] Beyond Antibody, Bengaluru, India. [5] The Campbell Family Institute for Breast Cancer Research and Ontario Cancer Institute, University Health Network, Toronto, ON, Canada. [6] Hugo W. Moser Research Institute at Kennedy Krieger, Department of Neurology, The Johns Hopkins University School of Medicine, Baltimore, MD, USA. [7] The Wilmer Eye Institute, The Johns Hopkins University School of Medicine, Baltimore, MD, USA. [8] Pharma Research and Early Development, Roche Innovation Center, F. Hoffmann-La Roche, Ltd, Basel, Switzerland. [9] Present address: Kodiak Sciences, Palo Alto, CA, USA. Correspondence and requests for materials should be addressed to D.S. (email: Debasish@pitt.edu)

Age-related macular degeneration (AMD) is a complex and progressive degenerative eye disease involving multiple genetic and environmental factors, leading to severe loss of central vision[1]. The vast majority of patients suffer from early, dry AMD, and, about half of these patients will develop advanced disease within 10 years. Despite the growing need, no definitive treatment or prevention for early, dry AMD is available. Inflammation plays a key role in the pathogenesis of various age-related diseases, including AMD[2–4]. Dysregulation of the innate immune system is critical for the onset of AMD; complement has been implicated, activation of various cytokines/chemokines, and the NLRP3 inflammasome have been invoked as central to AMD pathogenesis[5,6]. Inflammatory cells like microglia, monocytes/macrophages, and tissue-resident T cells, also appear to contribute to AMD pathobiology[7]. However, a role for neutrophils in AMD remains largely unexplored. In addition, the molecular mechanisms involved in immune system activation and regulation in AMD, and in the assembly of the inflammation-signaling platform, remain unknown.

Neutrophils play a central role in the innate immune response[8,9]. Our recent study revealed increased infiltration of lipocalin-2 (LCN-2) positive neutrophils into the choroid and retina of early, dry AMD patients as compared to age-matched controls[10]. It is now accepted that neutrophil subtypes that migrate to affected sites play a significant role in disease pathogenesis[11]. LCN-2, a protein involved in innate immunity, has been shown to be markedly elevated in serum and tissues during inflammation[12]. We have previously shown that LCN-2 is significantly higher in RPE cells of the aging Cryba1 (gene encoding βA3/A1-crystallin) cKO (conditional knockout) mouse, although we found no difference in younger mice[13].

While the lack of a comprehensive animal model of AMD limits our understanding of cellular mechanisms in the critical early disease stages, the mouse has been the model organism most used to study AMD[14,15]. We recently developed a genetically engineered mouse model that exhibits a slow progressive early, dry AMD-like pathology associated with inefficient lysosomal clearance decreasing both autophagy and phagocytosis in the RPE[16,17]. In the Cryba1 cKO mouse, these impairments lead to RPE cell degeneration including loss of basal infoldings, prominent intracellular vacuoles, and undigested melanosomes, as well as sub-retinal lesions at the posterior pole, deposits between the RPE and Bruch's membrane, decreased electroretinogram (ERG) signals, and photoreceptor degeneration as the disease progresses[13,16]. Our mouse model exhibits a slowly progressive form of AMD-like pathology associated with a chronic inflammatory immune response as the mice age, allowing us to test our hypothesis that infiltrating neutrophils homing to the retina during disease progression contribute to pathogenesis in early, dry AMD.

We demonstrate elevated interferon−lambda (IFNλ) in the retinae of human AMD subjects and in the Cryba1 cKO mouse model. This high expression of IFNλ in AMD retina signals the transmigration of neutrophils from the circulation into the retina during early AMD, eventually leading to major pathological sequelae. To the best of our knowledge, mechanistic studies showing that neutrophils may be activated in early AMD by signaling through the IFNλ/LCN-2/Dab2/integrin β1 axis, have not been previously reported. In the mouse model, inhibition of AKT2 reduced homing of neutrophils to the retina, decreased IFNλ expression, and alleviated early RPE changes.

## Results

### Infiltration of neutrophils in AMD and in a mouse model. As in human AMD[10], Cryba1 cKO mice present with immune cell

infiltration into the retina with aging (Fig. 1a). Flow cytometry analysis for the entire retinal cell population from posterior eyecups was performed by gating for CD45$^{high}$CD11b$^+$ cells (monocytes, macrophages, and neutrophils). The relative number of neutrophils (cells positive for Ly6C$^{high}$Ly6G$^+$) among CD45$^{high}$CD11b$^+$ cells in the tissue was determined, by simultaneously labelling cells with appropriate antibodies (Fig. 1a), as previously described[18]. While not increased in 2-month-old Cryba1 cKO retina, by 4 months, when an AMD-like phenotype is apparent in this mouse model, CD45$^{high}$CD11b$^+$Ly6C$^{high}$Ly6G$^+$ neutrophils were increased nearly three-fold relative to Cryba1$^{fl/fl}$ control retinas, and continued to increase with age, as seen in the 13-month-old Cryba1 cKO retina with respect to aged control mice (Fig. 1a). Furthermore, immunofluorescent analysis of retinal flatmounts from Cryba1 cKO mice confirmed an elevated number of Ly6G$^+$ cells in the retina relative to age-matched controls (Fig. 1b). An increase in sub-retinal neutrophils, as determined by Ly6G$^+$ staining of RPE flatmounts, was also observed in Cryba1 cKO mice relative to age-matched controls (Supplementary Fig. 1).

The percentage of neutrophils and their activation status in human early, dry AMD was studied by phenotyping the cells in peripheral blood (Supplementary Table 1) by flow cytometry using appropriate gating strategies (Supplementary Fig. 2). An increase in the proportion of CD66b$^+$ neutrophils within the total CD45$^+$ (leukocyte) population was observed in peripheral blood (Supplementary Fig. 3a) of AMD patients compared to control subjects. Further, an increased number of activated neutrophils (CD45$^+$CD66b$^{high}$) was observed in peripheral blood (Fig. 1c) with no change in the number of inactive neutrophils (CD45$^+$CD66b$^{low}$) (Supplementary Fig. 3b). We also observed a substantial increase in the total number of IFNλ receptor (IL-28R1)-positive leukocytes (CD45$^+$IL-28R1$^+$) in the peripheral blood of AMD patients (Fig. 1d). Moreover, IL-28R1$^+$ activated neutrophils (CD66b$^{high}$) were a markedly higher proportion of total neutrophils (CD66b$^+$ cells) in peripheral blood (Fig. 1e) from AMD subjects compared to age-matched controls. Immunolocalization studies show presence of CD66b$^+$ neutrophils in human tissue sections from normal and AMD samples (Supplementary Fig. 3c–g). We have previously shown that an increased number of neutrophils are present in the retina of human AMD patients compared to aged-matched control subjects[10]. However, IL28R1$^+$ expression is evident on CD66b$^+$ neutrophils only in retinal sections of AMD patients, but not in controls (Supplementary Fig. 3c–g), indicating that activated neutrophils home into the retina of only early AMD patients. These results indicate a greater propensity for IL-28R1$^+$ activated neutrophils to home into the eye, giving a probable scenario for the role of IFNλ-mediated signaling in these infiltrating neutrophils. It is known that once in the area of inflammation, neutrophils release Neutrophil Extracellular Traps (NETs), which can damage host tissue in immune-mediated diseases[19–22]. Indeed, early, dry AMD eyes showed increased staining for the NET markers, myeloperoxidase (MPO), neutrophil elastase and citrullinated histone H3 as compared to age-matched control eyes (Supplementary Fig. 4a–f). Taken together, our results support the idea that there is increased neutrophil infiltration into the retina during early, dry AMD.

### Factors promoting neutrophil infiltration into the retina.
RNAseq analysis was performed on retinal tissue obtained from 5- and 10-month-old Cryba1 cKO and floxed control mice in order to identify soluble factors, including cytokines and chemokines released from the retina, that may promote neutrophil infiltration. We found a major increase in the levels of IFNs, including IFNα, IFNγ, and IFNλ, as well as CXCL1 and CXCL9,

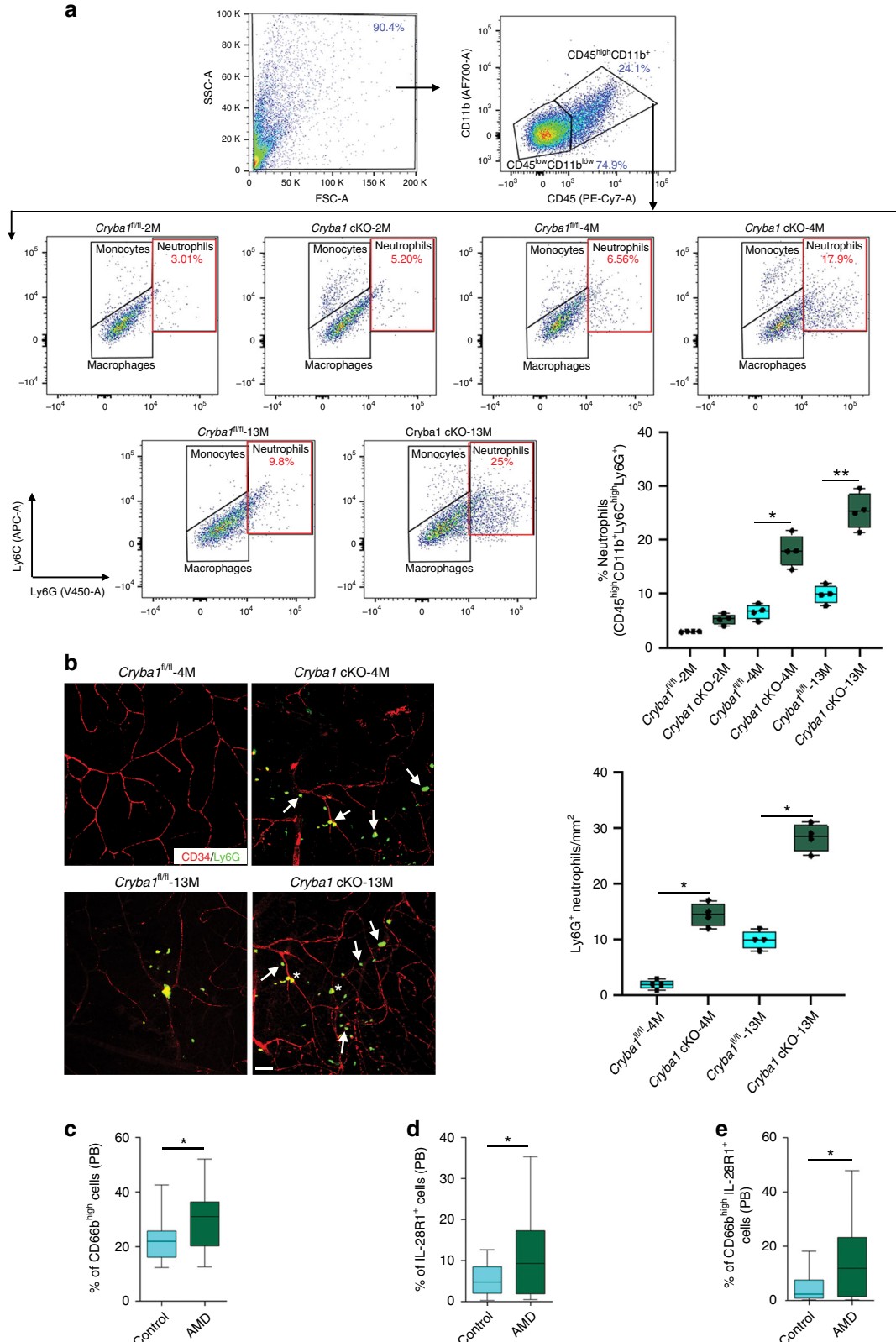

in the aged *Cryba1* cKO retinas compared to control (Supplementary Fig. 5). ELISA was performed to further confirm these results (Fig. 2a–c). Furthermore, to identify which cell types express IFNλ in the retina, immunofluorescence studies were conducted and showed a substantial increase in staining for IFNλ specifically in the RPE of AMD eye sections relative to

age-matched controls (Supplementary Fig. 6). Moreover, western analysis confirmed increased IFNλ and CXCL1 protein in human AMD RPE/choroid lysates, compared to control (Fig. 2d). In addition, we observed an increase in the levels of IFNα and IFNλ1 in the plasma and AH of early AMD patients compared to controls (Fig. 2e–h), but levels of IFNλ2/3 were not different in

**Fig. 1** Neutrophils accumulate into the retina of *Cryba1* cKO mice and in peripheral blood of human early AMD patients. **a** Representative dot plots are gated on the CD45+CD11b+ cells from mouse retina. The total population of CD45+CD11b+ cells is considered to be 100%, with CD45highCD11b+ (neutrophils, monocytes, and macrophages) and CD45lowCD11b+ (predominantly resident microglia) gated separately (arrows denoting population lineages). The level of Ly6C and Ly6G on the CD45highCD11b+ population was assessed to evaluate percentage neutrophils (%CD45highCD11b+Ly6ChighLy6G+ cells), which showed increased neutrophils only in 4- and 13-month-old *Cryba1* cKO mouse retinas compared to aged-matched controls (*Cryba1*fl/fl). No differences were observed between *Cryba1*fl/fl and cKO retinas at 2 months of age. $n = 4$. *$P < 0.05$ and **$P < 0.01$ (one-way ANOVA and Tukey's post hoc test). **b** Immunofluorescence studies and quantification of Ly6G+ cells (Green, Neutrophil marker) on retinal flatmounts, counterstained with CD34 (Red, marker for endothelial cells of blood vessels) revealed that neutrophils accumulated progressively in *Cryba1* cKO mouse retina (white arrows) and along the retinal blood vessels (yellow, asterisk), relative to age-matched control (*Cryba1*fl/fl). $n = 4$. *$P < 0.05$ (one-way ANOVA and Tukey's post hoc test). Scale bar, 50 μm. In early AMD patients, flow cytometry data revealed significant increase in the peripheral blood (PB) levels of **c** total neutrophils (CD66b+ cells), **d** total IL28R1+ cells and **e** IL28R1+ expressing activated neutrophils (CD66bhighIL28R1+). PB (AMD; $n = 43$ and Controls; $n = 18$). *$P < 0.05$ (Mann–Whitney test)

AMD patients compared to controls (Fig. 2i–j). The plasma levels of IFNγ showed substantial increase in AMD patients compared to control (Supplementary Fig. 7a), but no such change was found in the AH (Supplementary Fig. 7b). IFNβ and VEGF levels in the plasma and AH of AMD patients were not different from controls (Supplementary Fig. 7c–f). Thus, our results suggest a pro-inflammatory milieu in the eye, with a probable involvement of IFNλ, which is secreted from the diseased RPE thereby eliciting an inflammatory response. It is plausible that the increased levels of IFNλ might be the key factor that promotes the neutrophil activation and infiltration into the retina, since IFNλ receptor (IL28R1) is expressed on circulating neutrophils.

In addition to soluble factors, neutrophils also require adhesion molecules for their transmigration into the site of injury. Neutrophils adhere to endothelial cells when their integrins interact with endothelial cell immunoglobulin superfamily members[23], such as ICAM-1 and VCAM-1 (two important adhesion molecules on endothelial cells)[24,25], which enables them to transmigrate into diseased or injured tissue. We observed elevated levels of ICAM-1 (Fig. 2k) as well as VCAM-1 (Fig. 2l) in the retina of aged *Cryba1* cKO mice and human early, dry AMD patients respectively, relative to age-matched controls.

**IFNλ triggers LCN-2 expression and neutrophil activation**. It has been previously reported that IFNλ triggers phosphorylation and nuclear translocation (activation) of STAT1[26]. We have shown that during early AMD, STAT1 activation is critical for *lcn2* gene expression[10]. LCN-2 is an adipokine, known to be important for neutrophil activation and innate immune function[27]. In fact, we and others have shown that binding of NFκB and STAT1 to the promoter of LCN-2 causes pathogenicity[10,28]. Here, we show that mouse bone marrow-derived neutrophils cultured with either recombinant IFNλ or with conditioned medium from primary cultured RPE cells overexpressing IFNλ to simulate the increased IFNλ levels that we observe in the RPE of human AMD patients (Supplementary Fig. 6), exhibit increased levels of LCN-2 and phosphorylated STAT1 (Fig. 3a). Moreover, we also observed that IFNλ-exposed neutrophils showed a major increase in reactive oxygen species (ROS) levels (Fig. 3b) and phagocytosis (Fig. 3c). Increased formation of NETs was evident because of the prevalence of extracellular nuclear material (stained with DAPI), which showed increased staining for MPO and citrullinated Histone H3 (Fig. 3d), known markers of NETs[29]. Thus, the data suggests that IFNλ not only induces STAT1-mediated LCN-2 expression, but also potentiates neutrophil activation.

**LCN-2 activated neutrophils cause retinal degeneration**. We applied ribbon-scanning confocal microscopy (RSCM)[30] as a means to rapidly image red CMTPX-tagged neutrophils within an

entire NOD-SCID immune-deficient mouse eye to validate transmigration of activated neutrophils. The mice were intravenously injected with bone marrow-derived wild type (WT) neutrophils, bone marrow-derived neutrophils from LCN-2−/− (knockout) mice, WT neutrophils treated with IFNλ, or IFNλ treated neutrophils from LCN-2−/− mice. To demonstrate homing of activated neutrophils to specific regions of the eye, we performed RSCM paired with benzyl alcohol benzyl benzoate (BABB) clearing of NOD-SCID mouse eyes. The clearing procedure makes the refractive index consistent throughout the eye, thereby making the tissue transparent and allowing image acquisition throughout the depth of the whole organ. As shown in Fig. 4, NOD-SCID mice administered with red CMTX-tagged WT neutrophils showed little infiltration into the eye (Fig. 4a–d) and similarly, not many neutrophils derived from LCN-2−/− mice infiltrated the eye (Fig. 4e–h). The data clearly suggest that neutrophils home mostly into the choroid in both of these conditions, but due to the lack of stimuli from IFNλ in WT neutrophils and probably due to the perturbed migratory signaling axis in the LCN-2−/− mice, these cells fail to cross the intra-ocular compartments in considerable numbers through the blood-retinal or blood-aqueous barrier. Interestingly, red CMTPX-tagged neutrophils treated with IFNλ showed a noticeable number of neutrophils infiltrating the eye, mostly into the retina (Fig. 4i–l & Fig. 5d) relative to control (Fig. 4a–d & Fig. 5d). A 3D model shows the number and location of the infiltrating neutrophils in the eye (Fig. 5a–d and Supplementary Movie 1). We envisage that during early stages of AMD, neutrophils migrate from the peripheral blood into the intra-ocular compartments in response to a chemotactic cue, which we identified as IFNλ. In addition, NOD-SCID mice injected with IFNλ-treated LCN-2−/− neutrophils showed very few infiltrating cells into the retina (Fig. 4m–p) compared to mice injected with untreated LCN-2−/− neutrophils (Fig. 4e–h), demonstrating that neutrophil infiltration into the eye from the peripheral circulation is likely due to the IFNλ triggered LCN-2 activation.

To further validate our observations that increased LCN-2 levels induced by IFNλ in the transmigrating neutrophils can potentiate outer retinal degeneration, we injected NOD-SCID mice with bone marrow-derived WT neutrophils, bone marrow-derived neutrophils from LCN-2−/− mice, WT neutrophils treated with IFNλ, neutrophils treated with conditioned medium from primary cultures of RPE cells overexpressing IFNλ, or with recombinant LCN-2. After 7 days, Optical Coherence Tomography (OCT) analysis showed that mice injected with either IFNλ-treated WT neutrophils or recombinant LCN-2 exhibited alterations in the RPE and photoreceptor (inner and outer segments) layers (Fig. 6c–e). Quantitative analysis by spider plot revealed decreased thickness of these layers in the experimental groups (Fig. 6i and Supplementary Fig. 8a–c). No noticeable changes

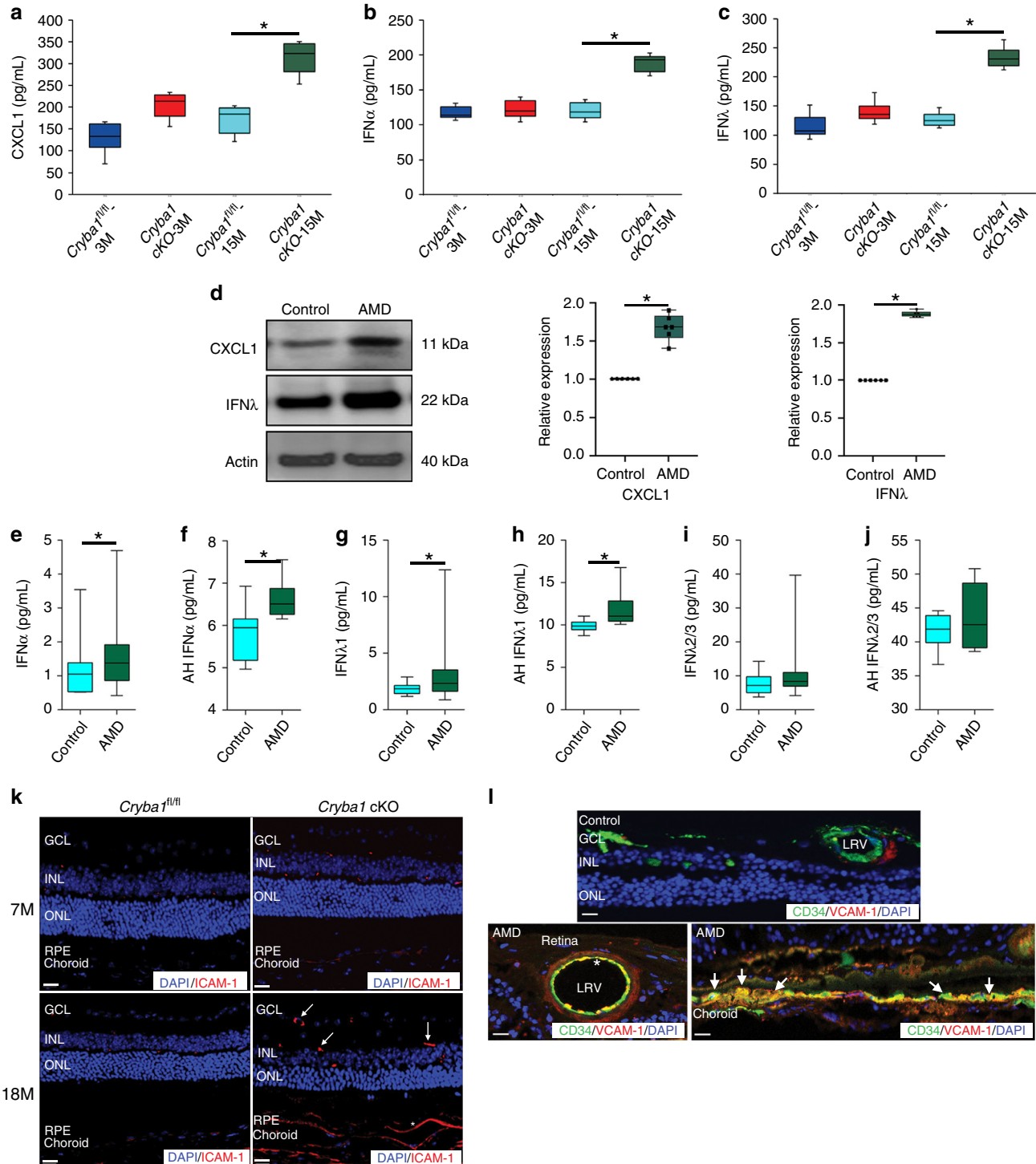

**Fig. 2** Increased levels of neutrophil regulating factors in retinas from *Cryba1* cKO mice and human early AMD donor eyes. The levels of **a** CXCL1, **b** IFNα, and **c** IFNλ were increased in the RPE-choroid tissue homogenate of 15-month-old *Cryba1* cKO mice compared to age-matched *Cryba1*fl/fl controls as measured by ELISA. No changes were observed in 3-month-old mice. $n = 6$. *$P < 0.05$ (one-way ANOVA and Tukey's post hoc test). **d** Representative immunoblot and densitometry showed elevated CXCL1 and IFNλ in RPE lysates from early AMD donor samples compared to age-matched controls. $n = 6$. *$P < 0.05$ (one-way ANOVA and Tukey's post hoc test). **e**–**j** Multiplex ELISA revealed significant increases in the levels of IFNα and IFNλ1 in plasma or aqueous humor (AH) of early AMD patients relative to controls. No noticeable change was observed in the plasma and AH levels of IFNλ2/3. Plasma (AMD; $n = 43$ and Controls; $n = 18$), AH (AMD; $n = 6$ and Controls; $n = 7$). *$P < 0.05$ (Mann–Whitney test). **k** Immunofluorescence assay showing increased staining of ICAM-1 (Red, neutrophil adhesion molecule) in the neural retina (white arrows) and RPE/choroid (asterisk) of aged (18-month-old) *Cryba1* cKO mice compared to age-matched control. No noticeable increase in staining was observed in the retina of 7-month-old *Cryba1* cKO mice. $n = 5$. Scale bar, 50 μm. **l** Immunostaining of human early AMD sections revealed increased staining of VCAM-1 (Red, neutrophil adhesion marker) in the large retinal vessels (LRV, asterisk), which were stained with CD34 (Green, marker for endothelial cells of blood vessels). Intense staining was also observed in the RPE/choroid (Yellow, white arrows). No noticeable staining for VCAM-1 was observed in the control sections. $n = 5$. Scale bar, 50 μm

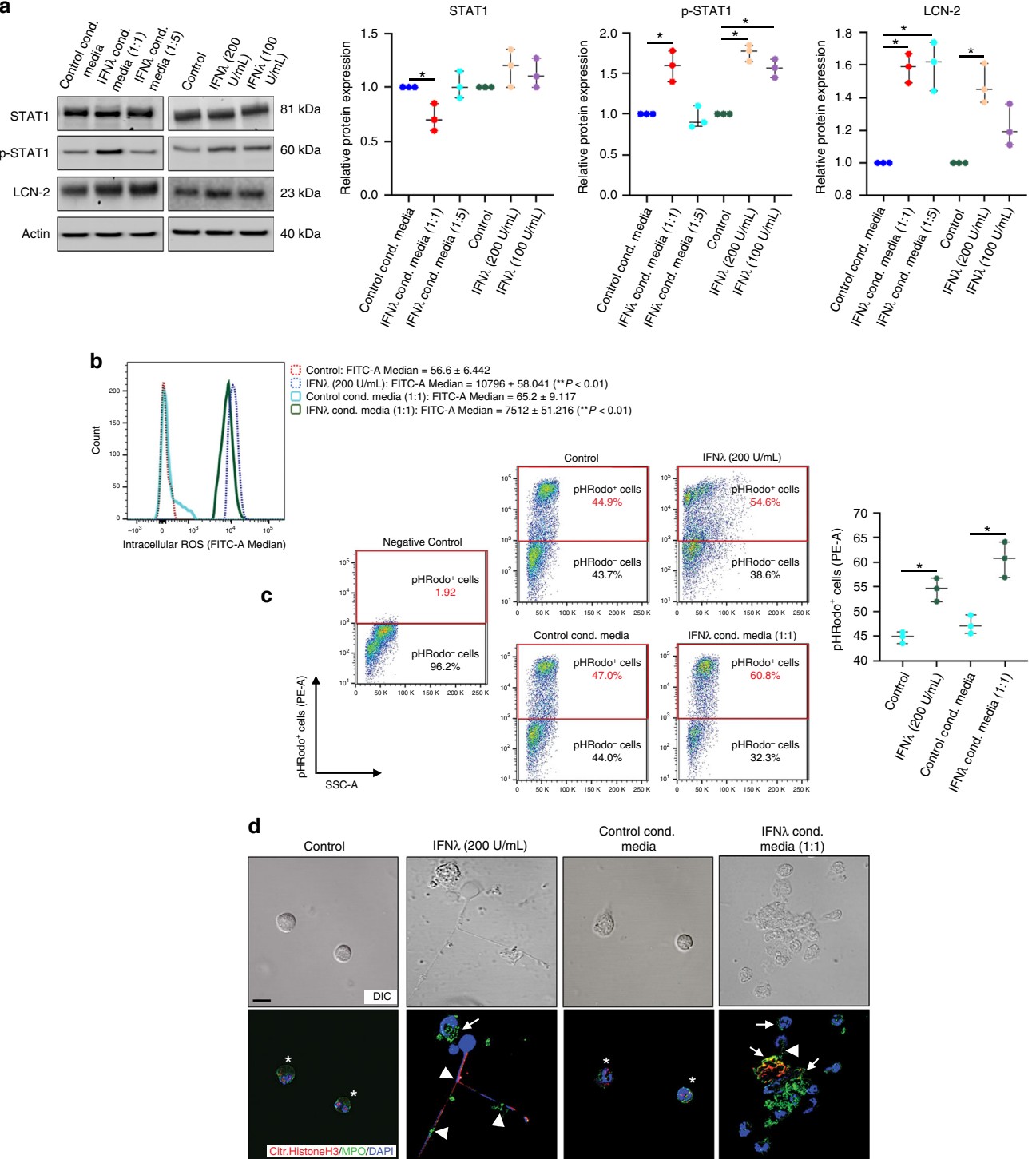

**Fig. 3** IFNλ triggers LCN-2 upregulation and activation in neutrophils. **a** Neutrophils exposed to conditioned media from IFNλ overexpressing RPE cells (for 6 h) or to recombinant IFNλ (for 2 h), showed increased expression of LCN-2 and p-STAT1 compared to control cells. IFNλ conditioned media (1:1) and recombinant IFNλ (200 U/mL) were used as the effective dose in other experiments, since they showed maximum effect in terms of LCN-2 upregulation. $n = 3$. *$P < 0.05$ (one-way ANOVA and Tukey's post hoc test). **b** Flow cytometric evaluation of intracellular ROS was performed by staining neutrophils from culture (as described in **a**) with 2',7' –dichlorofluorescein diacetate (DCFDA). ROS levels was represented by fluorescence intensity (FITC-A Median) values for 2',7'-dichlorofluorescein, (DCF, oxidized DCFDA), which showed significant increase in ROS levels among IFNλ-exposed neutrophils with respect to control. $n = 4$. **$P < 0.01$, with respect to control (one-way ANOVA and Tukey's post hoc test). **c** Phagocytosis assay was performed using pHRodo fluorescent labelled *E. coli*. particles in cultured neutrophils (as described in **a**). Flow cytometry analysis, upon gating on the negative control revealed, increased population of cells (red gating box), that have phagocytosed pHRodo *E. coli* conjugates among the IFNλ-exposed neutrophils groups relative to controls. $n = 4$. *$P < 0.05$ (one-way ANOVA and Tukey's post hoc test). **d** Neutrophil extracellular traps (NETs) were evaluated by staining cultured neutrophils (as described in **a**) with citrullinated histone H3 (Citr. Histone H3, Red) and myeloperoxidase (MPO, Green) antibodies. Increased double staining for NETs, which are extracellular nuclear material (DAPI, Blue), with MPO (Yellow, arrow heads) or with citrullinated histone H3 (Magenta, arrow heads) were observed in IFNλ-treated neutrophils. This was concomitant with increased cellular expression of MPO (arrow) in these cells. Controls did not show any extracellular nuclear material or NETs (asterisks). $n = 3$. Scale bar, 50 μm

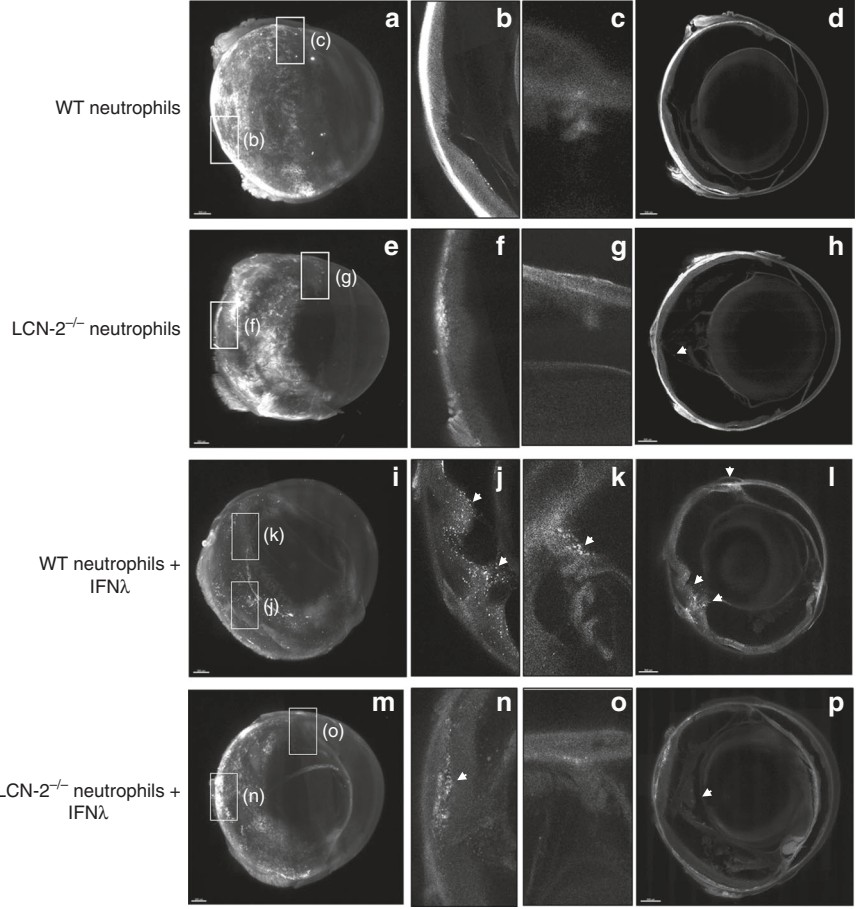

**Fig. 4** IFNλ triggers neutrophil homing into the eye in vivo. Ribbon scanning confocal microscopy (RSCM) was used to image neutrophil infiltration into whole cleared eyes from NOD-SCID mice intravenously injected with; untreated WT and LCN-2$^{-/-}$ neutrophils or IFNλ-exposed (200 U/mL), WT or LCN-2$^{-/-}$ neutrophils, tagged with red CMTPX. **a** 3D volumetric and **d** orthogonal projections from whole eyes obtained from mice injected with, WT neutrophils, did not show neutrophil homing **b** into the retina or **c** in through the aqueous humor drainage route (Schlemm's canal, a channel at the limbus and forms the joining point between the cornea and sclera, encircling the cornea). Mice injected with LCN-2$^{-/-}$ neutrophils showed **h** prevalence of neutrophils in the eye (arrow), but no infiltration was noticed into the **e**, **f** retina or **e**, **g** Schlemm's canal. Mice injected with IFNλ-treated WT neutrophils showed noticeable infiltration of neutrophils into the **i**, **l** eye (arrows), particularly in the **j** retina (arrow) and **k** Schlemm's canal (arrow), relative to untreated WT neutrophils. NOD-SCID mice injected with IFNλ-exposed LCN-2$^{-/-}$ neutrophils showed relatively lower numbers of neutrophils in the eye (arrow) (**m**, **p**), with respect to IFNλ-treated WT neutrophils, especially in the **n** retina (arrow). There was no visible neutrophil infiltration into **o** Schlemm's canal. $n = 1$. Scale bar, 300 μm

were observed in mice treated with vehicle and/or WT neutrophils (Fig. 6a, b and i). In addition, LCN-2$^{-/-}$ neutrophils, as well as LCN-2$^{-/-}$ neutrophils treated with IFNλ, showed no degenerative changes (Fig. 6f–i), suggesting a pathogenic role of LCN-2 in retinal degeneration. Hematoxylin-eosin staining of retinal sections from NOD-SCID mice, injected with either IFNλ-treated WT neutrophils or recombinant LCN-2, showed degenerative changes in the outer nuclear layer (ONL) along with photoreceptor layer (disruption of the inner and outer segments [IS/OS] junction) and RPE-Bruch's membrane-choriocapillaris complex (Fig. 6l–m), relative to vehicle control or WT neutrophil injected mice (Fig. 6j–k). Further, a thickness measurement of the retinal layers from these sections by spider plot showed severe loss or thinning of IS/OS and RPE layers in mice injected with either IFNλ-treated WT neutrophils or recombinant LCN-2, relative to vehicle or WT neutrophil-treated groups (Fig. 6r). In addition, immunofluorescence studies confirm increased photoreceptor and RPE cell loss in these mice, as evident from reduced staining for rhodopsin (labels rod photoreceptors) and RPE 65 (retinal pigment epithelium-specific 65 kDa protein) in

the retina of NOD-SCID mice, injected with either IFNλ-treated WT neutrophils or recombinant LCN-2 (Supplementary Fig. 8f–h), with respect to controls (Supplementary Fig. 8d–e). Therefore, our NOD-SCID mouse data provides novel evidence that IFNλ triggers LCN-2 activation in neutrophils, thereby inducing transmigration into the retina and potentiating retinal degeneration.

**Association of LCN-2/Dab2 regulates neutrophil infiltration.** These observations prompted us to further investigate the possible molecular mechanisms by which neutrophils infiltrate into the retina and thereby contribute to the pathogenesis of AMD. It has previously been shown that LCN-2 regulates neutrophil chemotaxis and cell migration in cancer cells[27,31]. To ascertain if IFNλ-mediated LCN-2 activation in neutrophils contributes to the increased adhesion and transmigration, we performed a human proteome high-throughput array to identify LCN-2 binding partners that may play a specific role in cell adhesion and migration. We found that LCN-2 interacts with Dab2 (Supplementary Fig. 9). This was confirmed by a pull-down assay,

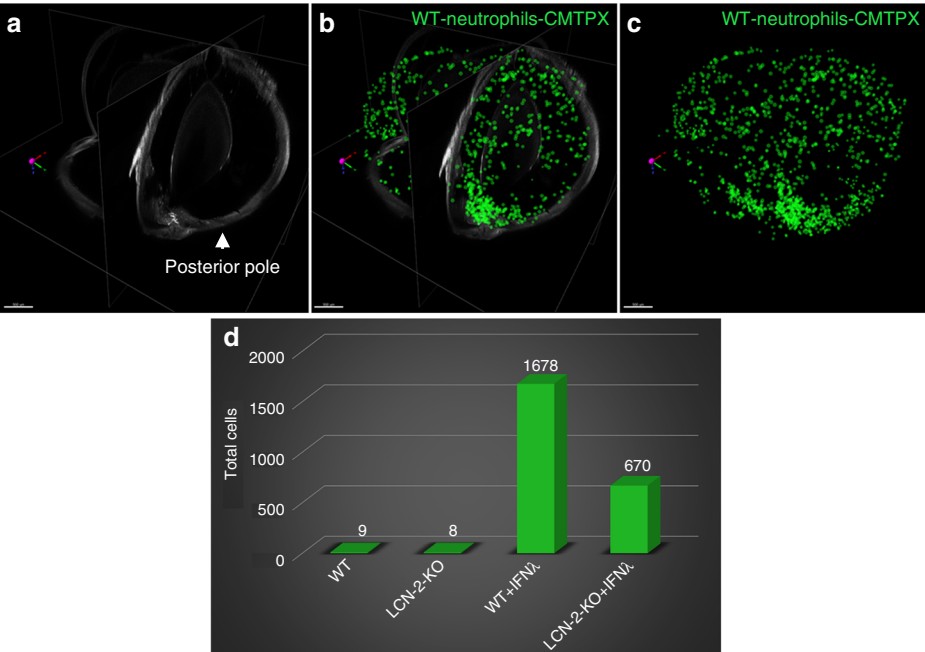

**Fig. 5** LCN-2 is responsible for neutrophil sequestration into the eye. **a** Orthogonal projections from all three dimensions of a whole eye from a mouse injected with WT neutrophils + IFNλ. Cells within the retina and Schlemm's canal were extracted as spot counts in Imaris software. Cells are depicted as green spheres **b** with and **c** without the orthogonal projection. **d** Counts extracted from all groups demonstrated an increase in neutrophil number (cell count) in the NOD-SCID mice injected (intravenous) with IFNλ-treated WT neutrophils compared to untreated controls, whereas loss of LCN-2 in neutrophils (LCN-2$^{-/-}$) showed reduced infiltration even after IFNλ exposure. $n = 1$. Scale bar, 500 μm

which showed an increased association between LCN-2 and Dab2 in IFNλ-exposed neutrophils as compared to untreated or control conditioned media treated neutrophils (Fig. 7a). It has previously been reported that, Dab2 binds to integrin β1 and regulates its internalization, thereby modulating cell migration[32]. It is also known that Dab2 is a negative regulator of cell adhesion particularly during inflammation[33,34]. Moreover, extracellular integrin β1 expression drives cell adhesion on the endothelial cell surface in various tissues thereby facilitating transmigration into the tissue[35,36]. We hypothesized that this increased association between LCN-2 and Dab2 may regulate extracellular integrin β1 level by modulating the Dab2/integrin β1 axis, thereby promoting neutrophil adhesion and transmigration into the retina. To explore the novel role of LCN-2 we used bone marrow-derived neutrophils from WT and LCN-2$^{-/-}$ mice that were cultured with either recombinant IFNλ or conditioned medium from IFNλ overexpressing RPE cells. Flow cytometry studies revealed an increase in extracellular integrin β1 expression in IFNλ-exposed neutrophils from wild type mice (Fig. 7b, c) concomitant with decreased integrin β1 (Fig. 7d). In addition, our co-immunoprecipitation data did not show any change in the binding between Dab2 and integrin β1 upon IFNλ exposure with respect to controls (Supplementary Fig. 10).These results suggest an alteration in the Dab2-mediated cellular internalization of integrin β1 in the IFNλ-exposed neutrophils, particularly due to the increased association between LCN-2 and Dab2 in the IFNλ-exposed cells (Fig. 7a).

Since we also observed neutrophils homing into the eye in NOD-SCID mice that were injected with IFNλ-exposed LCN-2$^{-/-}$ neutrophils (Fig. 5a), it is likely that the expression of adhesion-associated surface proteins is downregulated in the absence of LCN-2, as has been shown previously[37]. Based on these observations, we postulate that LCN-2 regulates the expression of extracellular adhesion molecules, which in turn modulates cell adhesion and

transmigration. However, there could be involvement of putative redundant pathways in regulating neutrophil infiltration upon exposure to IFNλ. We observed intensified neutrophil adhesion on fibrinogen-coated plates (Fig. 7e) and transmigration of IFNλ-treated normal neutrophils across fibrinogen-coated transwell chambers (Fig. 7f). In addition, we found that there is an increase in the extracellular expression of integrin β1 on untreated neutrophils from LCN-2$^{-/-}$ mice (Fig. 7b, c). This data is in sharp contrast to our previous observation that LCN-2$^{-/-}$ neutrophils treated with IFNλ has decreased surface expression of integrin β1 (Fig. 7b–c). Previous studies have shown that integrin β1 surface expression in neutrophils can be modulated by a number of independent signaling cascades during inflammation[38–41]. It is therefore plausible that integrin β1 in untreated LCN-2$^{-/-}$ neutrophils is upregulated independently of IFNλ/LCN-2/Dab2 pathway. But, the extracellular integrin β1 level and its internalization were stabilized in these LCN-2$^{-/-}$ neutrophils, even after IFNλ treatment, relative to IFNλ-exposed WT neutrophils (Fig. 7b–d). However, the adhesion and transmigration properties were greatly reduced in LCN-2$^{-/-}$ neutrophils exposed to IFNλ (Fig. 7e–f) and in integrin β1 silenced normal neutrophils (Fig. 7e–f), with no change in cell viability (Supplementary Fig. 11). These results suggest that LCN-2 regulates Dab2-mediated internalization of integrin β1, which is critical for cell adhesion and migration of IFNλ-exposed neutrophils.

**AKT2 inhibitor reduces AMD-like phenotype in *Cryba1* cKO mice.** We previously reported that AKT2 is an upstream regulator of NFκB-dependent *lcn2* gene expression[10]. Also, AKT2 can activate NFκB, which in turn is known to activate IFNλ and its downstream genes[42,43]. Therefore, we next asked whether CCT128930, a potent and selective inhibitor of AKT2[44], could block neutrophil infiltration into the retina by reducing the

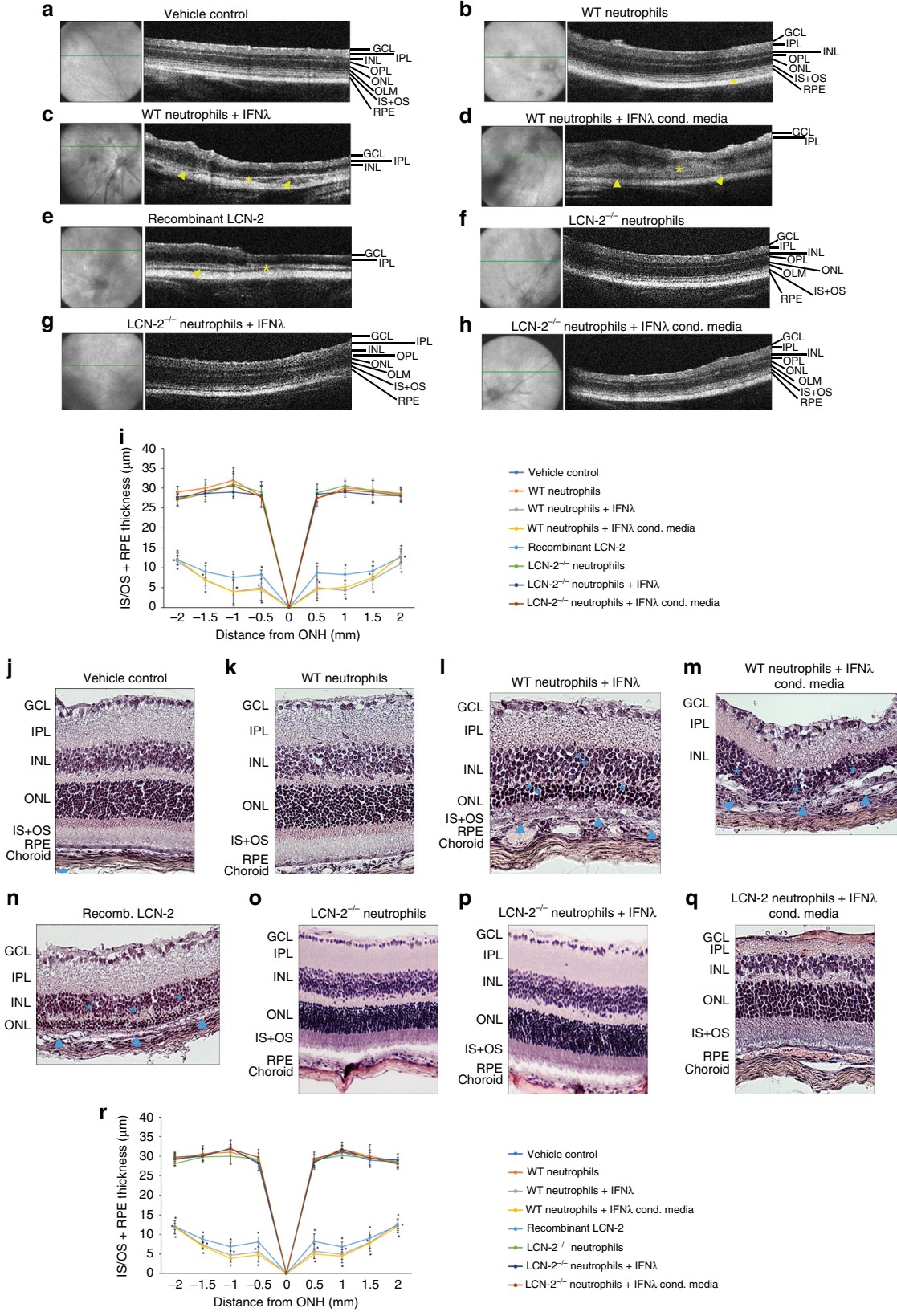

pro-inflammatory signal in the diseased retina. In the *Cryba1* cKO mice, the RPE is mildly degenerated at 12 months of age, progressing to severe RPE degeneration with photoreceptor degeneration by 20 months[16]. One-year-old *Cryba1* cKO mice

injected intravitreally with CCT128930 showed decreased expression of pAKT2, IFNλ and CXCL1 levels (Supplementary Fig. 12) in the RPE/choroid compared to the vehicle control. We also observed substantially fewer neutrophils in the retinas of

**Fig. 6** LCN-2 laden neutrophils promote AMD-like pathology. Representative spectral-OCT images of retinas from NOD-SCID mice injected sub-retinally with **a** vehicle (HBSS) or **b** WT neutrophils revealed normal retinal structure. In contrast, mice injected with; WT neutrophils pre-treated with either **c** recombinant IFNλ (200 U/mL), **d** conditioned media from IFNλ overexpressing mouse RPE cells (1:1 diluted) or **e** recombinant LCN-2 (10 pg/mL), show apparent changes in the ONL and INL layers (asterisks), concomitant with severe loss of RPE and IS + OS layer (yellow arrow heads). These alterations were not observed in mice injected with; **f** untreated neutrophils from LCN-2$^{-/-}$ mice or **g**, **h** IFNλ-exposed LCN-2$^{-/-}$ neutrophils. **i** Representative spider plot showing the thickness of the IS/OS + RPE layers using the OCT images among the experimental groups. $n = 10$. $^*P < 0.05$ (one-way ANOVA and Tukey's post hoc test). Hematoxylin-eosin staining showed no noticeable alterations in; **j** vehicle treated or mice injected with untreated **k** WT or **l–q** LCN-2$^{-/-}$ neutrophils (±) IFNλ. But, significant alterations were observed in the INL or ONL (blue asterisks) and RPE/IS + OS (blue arrow heads), in NOD-SCID mice sub-retinally injected with; **l**, **m** IFNλ-exposed WT neutrophils or **n** recombinant LCN-2. **r** Representative spider plot from all of the experimental groups showing the thickness of the IS/OS + RPE layers using the H&E images. $n = 5$. $^*P < 0.05$ (one−way ANOVA and Tukey's post hoc test), Scale Bar, 20 μm

CCT128930-treated cKO mice relative to those given vehicle only (Fig. 8a). Importantly, CCT128930 also reversed the early RPE degeneration and reduced the formation of deposits between Bruch's membrane and RPE (Fig. 8b–e). We have previously shown activation of Müller glia in our mouse model[10]. This condition, associated with reactive gliosis, is critical for the onset of the inflammatory process in most retinal diseases[45–47]. Interestingly, CCT128930-treated *Cryba1* cKO mice also showed considerable restoration of normal GFAP/CRALBP (Müller cell marker) staining relative to the vehicle-treated group (Fig. 8f). It is plausible that these changes may be linked to the reduction in the pro-inflammatory state in the retina of the CCT128930-treated cKO mice, as evident from decrease in neutrophil infiltration (Fig. 8a) and pro-inflammatory mediators like IFNλ and CXCL1 (Supplementary Fig. 12). As depicted in the schematic (Fig. 8g), our findings suggest that targeting the homing of activated neutrophils into the retina by specifically inhibiting AKT2-driven inflammation is potentially a novel therapeutic approach in early, dry AMD.

## Discussion

AMD is one of the leading causes of blindness in the elderly and is an immense socio-economic burden on the aging population. The dry or atrophic form comprises about 90% of all AMD cases, and no definitive treatment or prevention is available for these patients[48]. To uncover the cellular and molecular mechanisms involved in immune system activation and regulation in AMD, we examined aspects of early, dry AMD in the following: human AMD patient samples, a mouse model with an early, dry AMD-like phenotype (the *Cryba1* cKO)[16], NOD-SCID immunodeficient mice and LCN-2$^{-/-}$ mice. Using these tools, we show that IFNλ, a Type-III interferon, provides a signal for neutrophil homing into the retina during early AMD, by specifically upregulating LCN-2 in the neutrophils through the STAT1 pathway. We provide convincing evidence that LCN-2 regulates integrin β1-dependent neutrophil adhesion and transmigration. Increased expression of extracellular integrin β1 is known to increase cell adhesion, a requirement for increased transmigration of neutrophils[49]. We envisage that increased association between LCN-2 and Dab2 decreases integrin β1 internalization, which in turn increases the extracellular level of the integrin, activating transmigration into the retina and potentiating retinal degeneration.

Involvement of neutrophils in the pathogenesis of age-related diseases, such as Alzheimer's, and to some extent wet/neovascular AMD, has been previously reported[50,51]. In our previous study, we showed increased infiltration of LCN-2 positive neutrophils in the choroid and retina of early AMD patients compared to age-matched controls[10]. In addition to increased numbers of neutrophils in the retina, we found increased levels of activated neutrophils in the peripheral blood of AMD patients compared to age-matched controls. Increased IFNλ1 in the plasma and aqueous humor supports a scenario where IFNλ1 is associated with

increased activation of the surveilling neutrophils, possibly producing more inflammatory factors and engaging a feed-forward loop that stimulates disease progression. Since neutrophils typically have a short half-life, how do they contribute to AMD lesion formation? We suggest that chronic exposure to molecular triggers will repeatedly activate surveilling neutrophils, and if this pattern persists over time, the repeated inflammatory insult will contribute to tissue injury during AMD development. The previous reports that human neutrophils move into an activated state (CD66b$^{high}$) during inflammation and tissue infiltration are consistent with such a scenario[52].

To further substantiate our premise that homing of neutrophils into the retina with abnormal levels of LCN-2 potentiates outer retinal degeneration and aggravates RPE changes characteristic of early atrophic AMD[53], we injected NOD-SCID immunodeficient mice with WT and activated neutrophils. As expected, the data clearly showed that IFNλ-exposed activated neutrophils transmigrated into the retina and potentiated retinal degeneration. However, WT neutrophils or LCN-2$^{-/-}$ neutrophils that have lower levels of extracellular integrin β1, even after IFNλ treatment, failed to cause such an effect, strongly suggesting that abnormal levels of LCN-2 released from the infiltrating neutrophils trigger retinal degeneration. These data clearly corroborate our high-resolution RSCM imaging data illustrating the extravasation of large numbers of IFNλ-activated wild type neutrophils into the retina. The migration of neutrophils from the circulation to the site of inflammation is very well recognized[54]. However, we hereby propose a novel molecular mechanism directing the trafficking of neutrophils from the systemic circulation into the eye that results in retinal injury. Our findings suggest strongly that such a mechanism contributes to AMD progression. We believe that this process could be specific to the early stages of the disease and therefore a potential target for the development of novel treatments.

Taken together, we provide novel evidence that IFNλ triggers transmigration of neutrophils into the retina through activation of the LCN-2/Dab2/integrin β1 signaling axis leading to pathology in early AMD patients, as well as in a mouse model that mimics an early AMD-like phenotype[16]. Further, our findings suggest that targeting activated neutrophils by inhibiting AKT2 reduces neutrophil infiltration into the retina and reverses early AMD-like phenotype changes. We recognize that AKT2 inhibition can have other beneficial effects aside from reducing neutrophil infiltration, such as reducing activation of Müller glia, which could reduce or prevent AMD lesion formation. While antioxidant micronutrients slow intermediate AMD progression and anti-VEGF injections treat neovascular disease[55–57], no therapy is available for the earliest stages of the disease. Thus, AKT inhibitors should be assessed as potential therapy at the earliest stages of AMD. Several drugs targeting various isoforms of AKT are currently in different phases of clinical trials[58,59]. However, accumulating reports suggest adverse effects accompany treatment with AKT inhibitors. Therefore, understanding

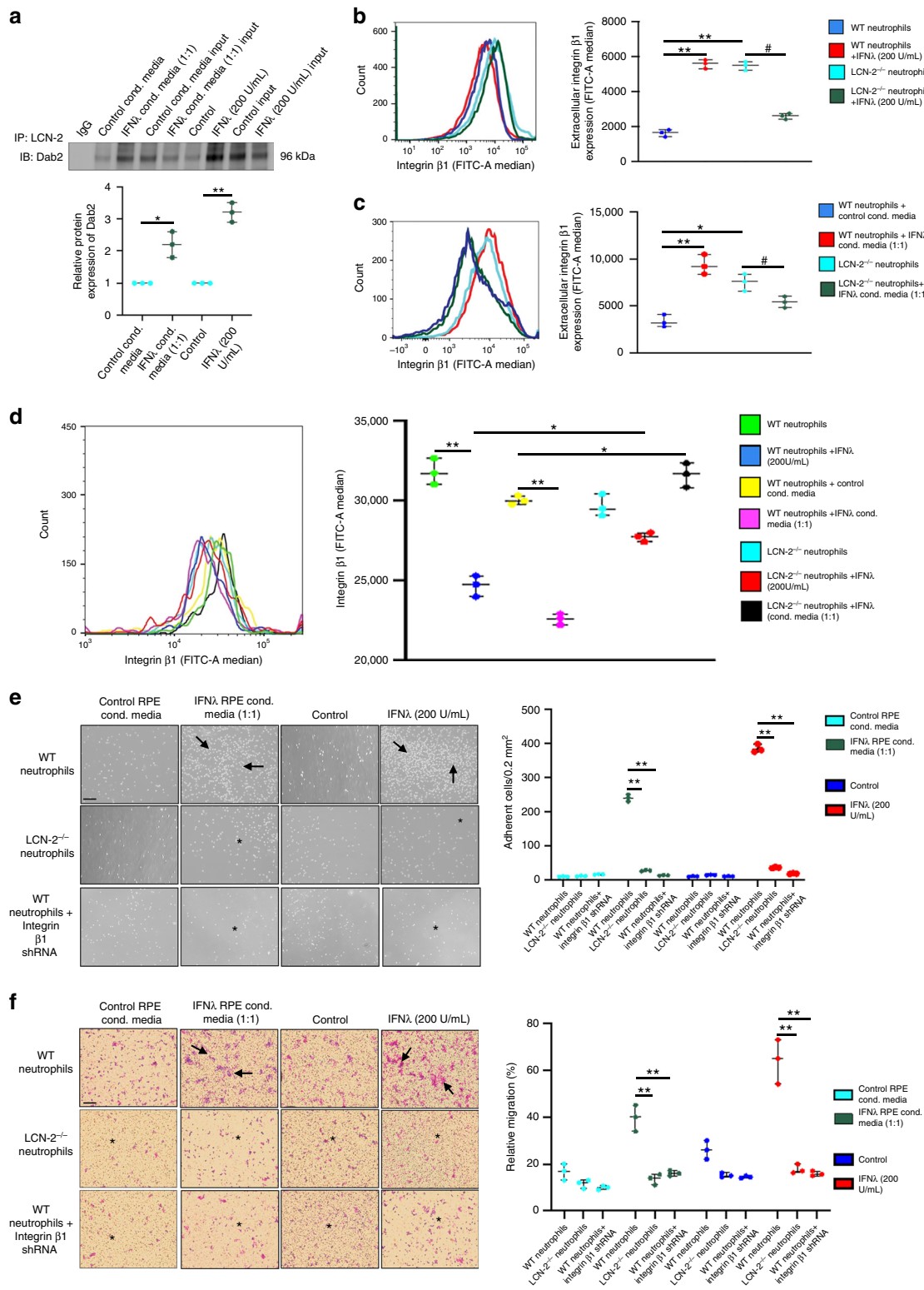

the consequences of localized inhibition in vivo as reported in this study might help to determine a dose of the inhibitor that could be effective without the side-effects, in particular diarrhea, hyperglycaemia and liver injuries, which have been observed in previous clinical trials of AKT inhibitors[60–62]. In addition, since we have delineated the signaling axis that is activated in the early stages of AMD, targeting individual components of this pathway

may also be highly beneficial for therapy. While we analyzed the entire NOD-SCID BABB-cleared mouse eye by high-resolution RSCM, our data do not demonstrate the route of entry of the neutrophils into the eye or the time course of their activation. We speculate that the red CMTPX dye-labeled activated neutrophils transmigrate into the retina through the retinal capillaries that constitute the blood-ocular barrier, however detailed knowledge

**Fig. 7** LCN-2 regulates neutrophil adhesion and transmigration by modulating the Dab2/integrin β1 axis. **a** Pull down assay from neutrophils exposed to conditioned media from RPE cells overexpressing IFNλ (1:1, for 6 h) or to recombinant IFNλ (200 U/mL, for 2 h) showed increased association between LCN-2 and Dab2 upon IFNλ treatment, relative to control. n = 3. *P < 0.05 and **P < 0.01 (one-way ANOVA and Tukey's post hoc test). **b–d** Flow cytometry assay showed increased extracellular and decreased intracellular expression of integrin β1 (FITC-A Median) respectively, in WT neutrophils treated with either recombinant IFNλ (200 U/mL, 2 h) or conditioned media from RPE cells overexpressing IFNλ (1:1 diluted, 6 h), compared to controls. Absence of LCN-2 in neutrophils (LCN-2$^{-/-}$) led to a reversal in the expression of both extracellular and intracellular levels on integrin β1, even after IFNλ treatment, relative to WT neutrophils. n = 3. *P < 0.05, **P < 0.01, and #P < 0.05 (one-way ANOVA and Tukey's post hoc test). **e, f** WT and LCN-2$^{-/-}$ neutrophils exposed to IFNλ (recombinant 200 U/mL for 2 h or 1:1 diluted conditioned media from RPE cells overexpressing IFNλ for 6 h), showed rapid adhesion to fibrinogen (20 mg/mL) coated plates (top panel, arrows: graph denotes adherent cells, counted in 0.2 mm$^2$) and transmigration across fibrinogen (150 mg/mL) coated plates (bottom panel, arrows: graph denotes relative migration (%) of cells, representative of cell count at the bottom of the insert using a computer assisted cell counter system). Integrin β1 shRNA transfected and LCN-2$^{-/-}$ neutrophils do not show changes in adhesion and transmigration even after IFNλ exposure (asterisk) n = 3. *P < 0.05 and **P < 0.01 (one-way ANOVA and post hoc test). Scale bar, 50 µm

of the route of entry and the number of activated neutrophils transmigrating into the retina would provide a window of time for a better-targeted therapy. Nevertheless, the present study provides a unique perspective to early, dry AMD by identifying neutrophils as an important pathophysiologic cellular component in the disease onset and progression. Hence, targeting neutrophils at the early stages of the disease is a viable strategy for treating early, dry AMD.

## Methods
**Antibodies**. PE/Cy7-tagged CD45 (Cat# 103114), APC-tagged Ly6C (Cat# 128016), FITC-tagged CD66b (Cat# 555724), V450-tagged Ly6G (Cat# 560603), Alexa fluor 700-tagged CD11b (Cat# 557960), anti-human PE/Cy7-tagged CD45 (Cat# 560178), and Anti human CD34 antibody (Cat# 343602) were purchased from BD Biosciences, USA and anti-human PE-tagged IL-28AR antibody (Cat# 337804) was purchased from Biolegend, USA. Anti-Neutrophil Elastase (Cat# ab68672), anti-GRO alpha (CXCL1) (Cat# ab86436), anti-STAT1 (phosphor S727) (Cat# ab109461), anti-IL28 receptor alpha or IL28R1 (Cat# ab224395), anti-Histone H3 citrullinated (Cat# ab219407), VCAM1 (Cat# ab134047), CD34 (Cat# 8158) and IL28 + IL29 (Cat# ab191426) antibodies were purchased from Abcam, USA. Anti-ICAM-1 (Cat# SC-107), Anti-STAT1 (Cat# 9172T), anti-AKT (Cat# 4685S), anti-AKT2 (Cat# 2964S) and anti-DAB2 (Cat# 12906S) were purchased from Cell Signaling Technologies, USA. Other antibodies used include: Alexa fluor 488-tagged β1 Integrin (Santa Cruz Biotechnology, USA; Cat# sc-374429 AF488), Anti-IL-28A/IFNλ2 (Antibodies online, USA; Cat# ABIN357173), Anti-Ly6G (Antibodies online, USA; Cat# ABIN1854937), IL-29 antibody (Biorbyt, USA; Cat# orb6201), anti-IFNα (Thermo Fisher, USA; Cat# 221001), anti-Myeloperoxidase/MPO (R&D Systems, USA; Cat# AF3667-SP), anti-LCN-2 (EMD Milipore; Cat# AB2267) and anti-Actin (Sigma Aldrich, USA; Cat# A2066).

**Animals**. Both male and female βA3/A1-crystallin conditional knockout mice (Cryba1 cKO C57Bl/6 J mice) and LCN-2 KO C57Bl/6 J mice were generated as previously explained[13,63]. NOD-SCID mice (NOD.CB17-Prkdescid/J; 4–5-weeks-old) were purchased from The Jackson Laboratory, USA. All animal studies were conducted in accordance with the Guide for the Care and Use of Animals (National Academy Press) and were approved by the University of Pittsburgh Animal Care and Use Committee.

**Human eyes**. The diagnosis and classification of AMD in human donor eyes was done as previously described[10]. For immunostaining, human donor eyes were obtained from the National Disease Research Interchange (NDRI; Philadelphia, Pennsylvania, USA) within 12–35 h of death. Donor eyes from 5 subjects with early, dry AMD (age range 79–95 years; mean age 85.8 years) and three aged controls (age range 77–89 years; mean age 82.5 years), with no evidence of macular disease were studied[10]. The study adhered to the norms of the Declaration for Helsinki regarding research involving human tissue. For immunophenotyping and soluble factors quantification experiments in human peripheral blood and aqueous humor, samples were collected from human donors, reporting to Narayana Nethralaya, Bangalore, India. All subjects underwent an ophthalmic exam, including visual acuity testing and retinal examination. Early AMD patients were diagnosed by fundus imaging, Amsler grid test and OCT imaging when deemed necessary and classified as per the AREDS[64]. Subjects with co-existing glaucoma or any other degenerative retinal disorders were excluded. The control group consisted of individuals without any history of AMD, diabetes, cardiovascular disorders or retinal diseases. 4–6 mL blood samples were collected in EDTA tubes from 18 controls and 43 AMD subjects by venipuncture. Aqueous humor samples (~50 μL) were collected from subjects undergoing cataract surgery (n = 7 control, n = 6 AMD) by anterior chamber paracentesis under sterile conditions. Within this group, early AMD subjects, where surgery is not contra-indicated, were identified by the presence of drusen and RPE abnormalities characterized by pigmentary

changes in the retina in accordance with AREDS classification. The demographic characteristics of the cohorts are described in Supplementary Table 1. All collected samples were immediately stored in a biorepository until further processing. All patient samples and related clinical information were collected after obtaining approval by the Narayana Nethralaya Institutional Review Board (IRB) and with written, informed consent from patients.

**Immunostaining**. Freshly enucleated eyes were fixed in 2% paraformaldehyde (PFA) for 10 min and then the anterior parts (cornea, lens, and attached iris pigmented epithelium) were removed. The resulting posterior eyecups were fixed in 2% PFA for 1 h at room temperature either for cryosections or RPE/retina flat mount. For cryosections, the eyecups were dehydrated through gradient sucrose solutions and embedded in OCT and for RPE/retina flat mounts, tissues were removed after the eyecup was quartered like a petaloid structure. The resulting eyecup was further cut radially into eight pieces from the optic nerve head to the periphery[17]. Immunostaining on human/mouse retina sections or on retina/RPE flatmounts were performed by using appropriate primary antibody (1:100) and incubated at 4 °C overnight. The RPE/retinal flatmounts or human or mouse retina sections were washed with 1X TBS thrice and then stained with appropriate secondary antibodies (1:300) with 1 μg/mL DAPI (Sigma Aldrich,USA) in the dark at room temperature for 2 h. The tissue sections or flatmounts were washed 6 times with 1X TBS. The tissues were mounted on a cover slip with DAKO mounting agent and then visualized under a confocal microscope (Zeiss LSM710, Switzerland)[10,17].

**Soluble factors quantification**. Peripheral venous blood was obtained by veni-puncture (n = 43 AMD patients and n = 18 control subjects) and aqueous humor (AH) was collected by anterior chamber paracentesis in AMD patients (n = 6) and control subjects (n = 7) from subjects undergoing cataract surgery. The levels of IFNα, IFNβ, IFNγ, IFNλ1–3, VEGF, and CXCL1 were measured in plasma and AH by bead-based multiplex ELISA (BioLegend, Inc, USA) using a flow cytometer (BD FACS Canto II, FACS DIVA software, BD Biosciences, USA). The absolute concentration for each analyte was calculated based on the standard curve using LEGENDplex™ software (Biolegend, Inc, USA).

**Immunophenotyping**. Cells from peripheral blood (n = 43 AMD patients and n = 18 control subjects) were labeled using fluorochrome conjugated anti-human antibodies specific for leukocytes (APC-Cy7-tagged CD45), neutrophils (FITC-tagged CD66b) and IFNλ receptor (PE-tagged IL-28R1) at room temperature for 45 min. Red blood cells from peripheral blood samples were lysed in 1X BD lysis buffer for 10 min, washed and resuspended in 1X phosphate buffered saline prior to flow cytometry (BD FACS Canto II, FACS DIVA software, BD Biosciences, USA) based acquisition and analysis. Data were analyzed using FCS Express 6 Flow Research Edition software. The leukocyte populations were identified by manual gating using SSC/CD45$^+$ profile. Subsequent gating was done on SSC/CD66b FITC to identify neutrophils. The neutrophil activation status was determined based on CD66b cell surface expression. CD45$^+$CD66b$^{high}$ cells were considered as activated neutrophils and CD45$^+$CD66b$^{low}$ as inactive neutrophils. CD45$^+$CD66b$^{high/low}$ IL-28R1$^+$ indicated IFNλ receptor positive neutrophils. The percentage of positive cell events for each staining panel was calculated.

**RPE isolation and culture**. Mouse RPE was isolated from control C57BL/6 J mice (3-weeks-old, n = 9; Jackson Laboratories, USA) and cultured by enucleating the eyes and then washed twice in DMEM containing high glucose and incubated in 2% (weight/volume) Dispase (Roche, 10269638001) in DMEM for 45 min at 37 °C. The eyes were then washed twice in growth medium made of DMEM (high glucose) containing 10% FCS, 1% penicillin/ streptomycin, 2.5 mM L-glutamine, and 1X MEM nonessential amino acids (Gibco, Invitrogen, 11095). An incision was made around the ora serrata of each eye and the anterior segment was removed. The resulting posterior eyecups were placed in growth medium for 20 min at 37 °C to initiate separation of the neural retina

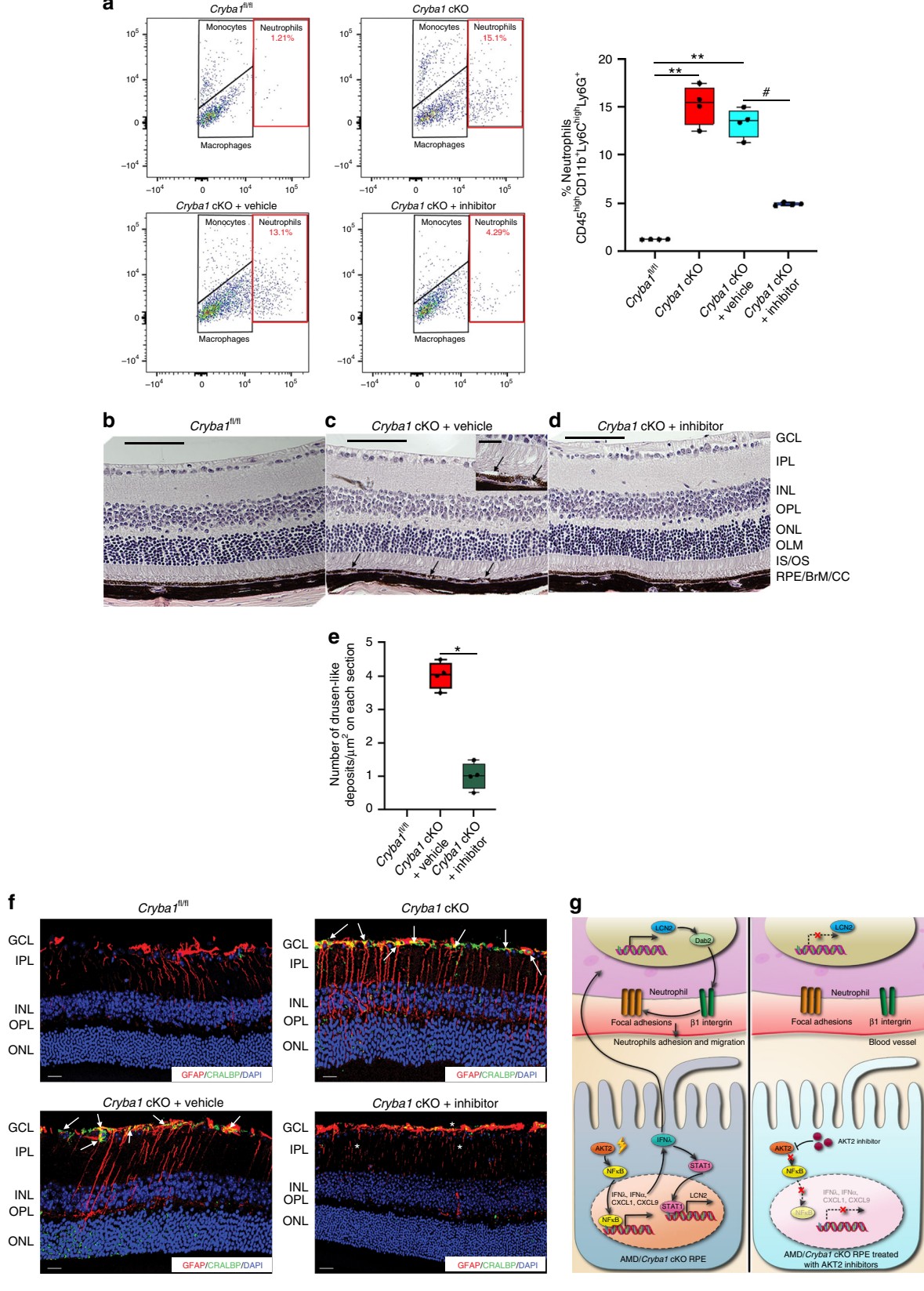

from the RPE. The neural retina was removed and intact sheets of RPE cells were peeled off the underlying Bruch's membrane and transferred in a sterile 60-mm culture dish, containing fresh growth medium. The RPE sheets were washed thrice with growth medium and then twice with calcium and magnesium free HBSS and then briefly triturated, using a fine point Pasteur pipette. RPE cells were centrifuged at 200 g for 5 min and cultured in transwell plates in growth medium[65].

**IFNλ overexpression in cultured RPE cells**. pLV-C-IL28A-GFPSpark and control vector were purchased from Sino Biological Inc. (Beijing, China, Cat# MG51305-ACGLN). Primary mouse RPE cells (in a monolayer; 90% confluent) were transfected with the respective vectors using X-tremeGENE transfection reagent (Roche, Switzerland) following the manufacturer's instructions. The transfection efficiency was estimated by evaluating the level of IL-28A/IFNλ released (into the cell-free supernatant) from overexpression transfected RPE cells by ELISA, with respect to

Fig. 8 Inhibiting AKT2 phosphorylation blocks neutrophil infiltration into the retina and rescues early RPE changes in *Cryba1* cKO mice. **a** Flow cytometry dot plots denoting monocytes, macrophages and neutrophils from mouse retina (as explained in Fig. 1a). The neutrophil population (%CD45$^{high}$CD11b$^+$ Ly6C$^{high}$Ly6G$^+$ cells, red gated) significantly increased in the 12 month *Cryba1* cKO mouse retina ± intravitreal vehicle treatment, compared to age-matched *Cryba1*$^{fl/fl}$ (control). Intravitreal treatment with the AKT2 inhibitor (CCT128930) significantly reduced neutrophils in cKO retina. Graphs denote % CD45$^{high}$CD11b$^+$Ly6C$^{high}$Ly6G$^+$ cells. $n = 4$. **$P < 0.01$ and $^\#P < 0.05$ (one-way ANOVA and post hoc test). **b–d** Representative histological sections (H&E) of retina from 1 year old *Cryba1*$^{fl/fl}$ mouse, showing normal structure (**b**). Age-matched *Cryba1* cKO mouse (**c**) intravitreally injected with vehicle (2.5% DMSO in PBS) shows RPE and photoreceptor lesions with pigmentation changes (arrows). Inset in **c**, shows higher magnification of RPE lesions indicating possible debris accumulation between Bruch's membrane and RPE and separation of photoreceptors from RPE (arrows). In contrast, inhibitor (CCT128930, inhibits AKT2 activation) treated *Cryba1* cKO mice (**d**), exhibited normal structure after 4 weeks. **e** Bar graph showing decrease in number of sub-retinal drusen-like deposits after AKT2 inhibitor treatment compared to vehicle-treated cKO mice. $n = 4$. Scale bars, 100 and 50 μm (inset). *$P < 0.05$ (one-way ANOVA and post hoc test). **f** Retina sections from 12-month-old *Cryba1*$^{fl/fl}$ or *Cryba1* cKO mice stained with glial fibrillary acidic protein (GFAP, red) and cellular retinaldehyde-binding protein (CRALBP, green). Sections from cKO mice ± intravitreal vehicle showed extensive staining of the Müller glial processes (cells staining for both CRALBP and GFAP, yellow indicating activation, arrows). This was significantly reduced after inhibitor treatment (asterisk). $n = 4$. Scale Bar, 50 μm. **g** Schematic depicting neutrophils homing into the retina and releasing LCN-2, generating pro-inflammatory conditions that contribute to elements of early AMD pathobiology. Our data suggest that IFNλ triggers transmigration of neutrophils into the retina through activation of the LCN-2/Dab2/integrin β1 signaling axis (Left panel). Inhibiting AKT2-dependent signaling can neutralize inflammatory signals and block neutrophil infiltration (Right Panel). Thus, AKT2 inhibitors should be assessed as potential therapy at the earliest stages of AMD

the control vector transfected cells; a minimum of a three-fold increase in IL-28A/ IFNλ level was considered appropriate for performing further experiments with the conditioned media.

**Isolation and culture of neutrophils**. Neutrophils from WT and LCN-2$^{-/-}$ mice were isolated by centrifugation of bone marrow cells, flushed from femurs and tibias and purified over a Percoll discontinuous density gradient following isolation, neutrophils were resuspended at a density of $10 \times 10^6$ per ml in Ca$^{2+}$ and Mg$^{2+}$ free HBSS, supplemented with 20 mM HEPES and then cultured in 37 °C at a density of $3 \times 10^6$ cells per ml before stimulation with either recombinant IFNλ or conditioned media from RPE cells overexpressing IFNλ[50,66].

**pHrodo phagocytic assay**. Neutrophils in culture were incubated with fluorescent-tagged particles (pHrodo™ Red E. coli BioParticles™ Conjugate for Phagocytosis assay kit, Thermo Fisher, USA, Cat# P35361) and flow cytometric evaluation of percentage cells which has engulfed the pHrodo particles (phagocytic cells) was performed by following the manufacturer's protocol.

**Integrin β1 shRNA transfection**. Integrin β1 shRNA lentiviral (Cat# sc-60044-V) and control shRNA (Cat# sc-108080) particles were purchased from Santa Cruz Biotechnology, USA. Mouse bone marrow-derived neutrophils ($5 \times 10^6$ cells/mL in HBSS containing 20 mM HEPES) were plated and then transfected with integrin β1 shRNA lentiviral or control shRNA particles for 8 h, according to the manufacturer's protocol.

**Rapid neutrophil adhesion assay**. Mouse bone marrow-derived neutrophils ($5 \times 10^6$ cells/mL in HBSS containing 20 mM HEPES) from LCN-2$^{-/-}$ mice and WT mice respectively or neutrophils transfected with either control shRNA or integrin β1 shRNA were subjected to rapid adhesion assay. Glass bottom 35 mm plates were coated for 16 h at 4 °C with human fibrinogen (20 μg/well in endotoxin-free PBS). Neutrophils from all experimental conditions ($10^5$ per well; $5 \times 10^6$ per mL in 10% FCS, 1 mM CaCl2/MgCl2 in PBS, pH 7.2) were added, incubated for 10 min at 37 °C, and then fixed on ice in 1.5% glutaraldehyde for 60 min and then counted with computer assisted enumeration[50].

**Neutrophil transmigration assay**. Neutrophils ($5 \times 10^6$ cells/mL in HBSS containing 20 mM HEPES medium) from LCN-2$^{-/-}$ and WT mice respectively or neutrophils transfected with either control shRNA or integrin β1 shRNA were used to assess cell migration by using transwell plates[50]. Neutrophils were plated on transwell inserts at $5 \times 10^6$ cells per ml and then exposed to different experimental conditions and cultured at 37 °C. The cells at the bottom of the transwell were fixed with 1.5% glutaraldehyde for 60 min, stained with Giemsa and then counted with computer assisted enumeration[50].

**Estimation of percentage neutrophils in mouse retina**. Mouse retinas were dissected from enucleated eyes and digested with 0.05% collagenase D (Roche, Switzerland, Cat# 11088858001) at 37 °C for 30 min, teased with blunt end forceps and pipetted to release cells, passed through a 70 μm cell strainer, and centrifuged at 1300 g, 4 °C for 20 min. The entire pellet was used for assessing the % neutrophils by flow cytometry, after staining with anti-Ly6G, Ly6C, CD11b, and CD45 antibodies at a concentration of 1 μg/mL for 90 min at room temperature[67].

**Intracellular ROS**. Flow cytometry was performed to evaluate the intracellular ROS in neutrophils by staining cells ($1 \times 10^6$ cells) from each experimental group with 2′,7′–dichlorofluorescin diacetate (DCFDA, Sigma Aldrich, USA, Cat# D6883–50MG) (25 μg/ml) for 30 min at 37 °C. Excess DCFDA was washed and cells were resuspended in PBS. The ROS content of the cells was measured on a flow cytometer[68].

**Estimation of intracellular and extracellular expression of integrin β1**. Freshly cultured bone marrow-derived neutrophils from WT and LCN-2$^{-/-}$ mice were incubated with Alexa fluor 488-tagged β1-Integrin (Santa Cruz Biotechnology, USA) antibodies at a concentration of 1 μg/mL in PBS containing 1% BSA for 1 h and the cell surface expression of integrin β1 (FITC fluorescence) was evaluated among these cells as described previously[32]. For intracellular expression of integrin β1, cells were permeabilized with 0.1% Triton X-100 in PBS for 5 min at 25 °C before incubating with anti-integrin β1 antibody at a concentration of 1 μg/mL in PBS containing 1% BSA for 1 h. Cell were analyzed by flow cytometry[68].

**SDS-PAGE and western blot analysis**. SDS-PAGE and western blot analyses were performed by suspending and sonicating cells or tissue samples in RIPA lysis buffer (Millipore, Billerica, MA, 20–188) plus 1% protease and phosphatase inhibitors (Sigma)[17]. Samples were placed on ice for 20 min and then centrifuged at 13,000 g for 20 min in 4 °C. The supernatants were subjected to protein estimation by BCA kit (Thermo Fisher, USA). Twelve micrograms of protein was used per sample and mixed with 4X protein sample buffer (Invitrogen, Carlsbad, CA) with 5% 2-mercaptoethanol (Sigma Aldrich, USA) and heated at 100 °C for 10 min. Samples were loaded into a 4–12% Bis-Tris Nu-PAGE gel (Invitrogen), electrophoresis was performed in MES buffer (Novex, Waltham, MA, USA). Proteins were transferred to nitrocellulose membranes and blocked with 5% skim milk (Biorad, USA) or 5% BSA (Sigma, for phosphorylated proteins)[17]. The primary antibodies were used at a dilution of 1:1000 whereas, all secondary antibodies were used at a dilution of 1:3000.

**Preparation of recombinant LCN-2 protein**. Full length LCN-2 cDNA was synthesized by GeneScipt, USA. It was subcloned in pET28a vector at NdeI and XhoI restriction site. The construct was transformed into *E.coli* DH5-α cells for amplification and *E.coli* Rosetta for expression. A single colony was grown overnight as a mother culture. 10% of mother culture was inoculated and grown to 0.8–1.0 OD and induced with 0.5 mM IPTG for 2 h at 37 °C. The cells were then pelleted by centrifugation at 6000 rpm for 10 min at 4 °C in a microfuge, resuspended in 10% volume of 20 mM Tris pH 8.0, containing 300 mM NaCl and 10% Glycerol. The mixture was sonicated for 30 s on and off each for 6 cycles, and then centrifuged at 12000 rpm for 30 min at 4 °C. The supernatant fraction was passed over a Nickel NTA (BioVision, USA) column as per the manufacturer's protocol. The column was washed twice with 10 times the bed volume with 20 mM Tris pH 8.0, with 300 mM NaCl, 10% Glycerol and 20 mM Imidazole. The protein was eluted with 20 mM Tris pH 8.0, 300 mM NaCl, 10% Glycerol and 300 mM Imidazole with approximately five times the bed volume in multiple fractions. The protein was polished over Sephacryl S-300 (GE Healthcare, USA, GE17–0599–10) following overnight dialysis at 4 °C in 1X PBS and 50% Glycerol. The filter (0.25 micron) sterilized protein was stored at −20 °C in working aliquots.

**Protein–protein interaction**. The human proteome microarray 2.0 analysis was performed as a paid service from CDI NextGen Proteomics, MD, USA. Recombinant LCN-2 was analyzed for protein–protein interaction profiling on the HuProtTM v3.1 human proteome array and the sample was probed on array plates at 1 μg/mL, with data analyzed using GenePix software. Hit identification was assessed as the ratio of median value of the foreground to the median of the surrounding background for

each protein probe on the microarray, followed by normalization to the median value of all neighboring probes within the $9 \times 9 \times 9$ window size and represented as the significance of the probe binding signal difference from random noise (Z-Score). The cutoff Z-score was 6 in this study for the triplicate analysis; only protein interactions with a Z-score above 6 were considered[17].

**Enzyme-linked immunosorbent assay**. The RPE choroid complexes harvested from freshly enucleated mouse eyes were kept on ice and then homogenized in 300 μL of complete extraction buffer (Abcam, USA, Cat# ab193970). The homogenized tissue was used to perform Enzyme-linked immunosorbent assay (ELISA) on 96-well microtiter plates coated with tissue lysates and incubated overnight at 4 °C. The plates were blocked with 5% BSA for 2 h. After washing, 50 μl of appropriate primary antibody, diluted to 1:1000 was added to each well and incubated for 2 h at room temperature. Bound cytokine was detected with secondary IgG peroxidase (Sigma Aldrich, USA). The color was developed with TMB substrate solution (BD Pharmingen, USA). The reaction was stopped with 2 N H2SO4 solution and absorbance was measured at 450 nm using a microplate reader[69].

**Clearing and imaging of whole eyes**. Whole mouse eyes harvested from animals that had been injected with red CMTPX labeled neutrophils were fixed overnight in 4% paraformaldehyde. As described previously[70,71], eyes were subject to clearing by BA:BB through a series of PBS:Ethanol gradients to dehydrate the organs prior to clearing with a 1:2 mixture of benzyl alcohol (Sigma, 305197) and benzyl benzoate (Sigma, B6630). After the samples were visibly clear, they were mounted in BA:BB solution between cover glass.

Each eye was scanned using the RS-G4 ribbon scanning confocal (Caliber ID) fitted with a 20 × /1.00 Glyc (CFI90 20XC, Nikon), correction collar set to 1.50. Linear interpolation of 561 nm laser excitation (iChrome-MLE-LFA, Toptica) was set between 15–30% power, top to bottom of z-stack. Emission was detected using a 630/69 band-pass filter, PMT settings were HV, 85; offset, 5. Voxels measured (0.395 × 0.395 × 5.33 μm). Each sample required ~3.5 h of total acquisition time. Imagery was collected at 16 bit pixel depth and comprised ~65 GB per eye. Images were collected as ribbons and were stitched and assembled using custom algorithms in MATLAB v2017b. Each dataset was converted to Nikon ND2 format and deconvolved with a custom NIS-Elements application configured for the Richardson-Lucy algorithm, line-scanning confocal, image noise level high and 0.76 μm pinhole. The deconvolved images were then converted to IMS format and loaded in to Imaris 9.2.1 (Bitplane). Prior to any analysis in Imaris, a Gaussian Filter was applied with a filter width of 0.395.

Neutrophils were quantified using the spot count function in Imaris Surpass. Spots were quantified over the entire image and then manually edited to maintain only those spots that were within the retina and Schlemm's canal. Finally, all remaining spots were filtered by volume to eliminate any structure that did not fall between 728–5800 μm³ (11–22 μm diameter), an approximate diameter of neutrophils, and to eliminate structures that were falsely selected during spot counting. Imaris surpass spot counting parameters were the same for all datasets: Enable Region Growing = true; Estimated Diameter = 10.0 μm; Background Subtraction = true; "Quality" above 190; Region Growing Type = Local Contrast; Region Growing Manual Threshold = 139.744; Region Growing Diameter = Diameter From Volume.

**RNAseq analysis**. RPE-Choroid from enucleated eyes harvested from 5- and 10-month-old *Cryba1*^fl/fl and *Cryba1* cKO mice (*n* = 4), respectively, were subjected to total RNA isolation as previously described[66]. Approximately 30 ng/μL total RNA was used to perform RNA-sequencing as a paid service from DNA Link, USA. All sequence reads were mapped to the reference genome (NCBI37/mm9) using the RNAseq mapping algorithm included in CLC Genomics Workbench. The maximum number of mismatches allowed for the mapping was set at 2. To estimate gene expression levels and analyze for differentially expressed genes among the different groups, RPKM was calculated[72].

**Co-Immunoprecipitation**. To evaluate the association between LCN-2/Dab2 and also Dab2/integrin β1 in different experimental conditions, cultured neutrophils from different experimental groups were subjected to co-immunoprecipitation (Co-IP) using the Pierce™ Co-Immunoprecipitation Kit (Thermo Fisher, USA, Cat# 26149). The cells were sonicated in IP Lysis/Wash Buffer (provided in the kit) plus 1% protease inhibitors (Sigma Aldrich, USA). The total lysates were processed with the kit according to the instructions. Seventy micrograms of lysates of each group were immunoprecipitated with 10 μg immobilized LCN-2 and dab2 antibodies respectively at 4 °C overnight. Normal rabbit IgG (Santa Cruz, USA) was used as the negative control. After elution, samples were loaded (15 μg per well) in SDS-PAGE and western blot was performed[17].

**Intravitreal injection of AKT2 inhibitor**. *Cryba1*^fl/fl and *Cryba1* cKO mice (Male, 12-months-old; *n* = 4) were intravitreally injected with 2 μl inhibitor (500 μM of CCT128930 in 2.5% DMSO in PBS) or vehicle only (2.5% DMSO in PBS) into the vitreous, once every week for three weeks. All instruments were sterilized with a steam autoclave. Bacitracin ophthalmic ointment was applied postoperatively.

Animals were euthanized with CO2 gas 4 weeks after the first injection and the retinas were harvested for further study[10].

**Sub-retinal injection of neutrophils in NOD-SCID mice and OCT**. NOD-SCID mice (NOD.CB17-Prkdescid/J, Jackson Laboratories, USA, male, 4–5-weeks-old) were used for the study. A large sample size, *n* = 10, was taken to nullify any experimental anomaly. Mice were anaesthetized and sub-retinal injections of neutrophils from different experimental groups or recombinant LCN-2 protein were given as described earlier[73]. Seven days after treatment, the NOD-SCID mice were anaesthetized by intraperitoneal injection of a ketamine and xylazine mixture and then subjected to Fundus imaging along with OCT analysis using the Bioptigen Envisu R2210 system. OCT images were analyzed on optical sections (100 sections per retina) from each eye ranging from−2.0 to +2.0 mm with respect to the optic nerve head (ONH) using the FIJI-ImageJ (NIH) plugin provided with the instrument along with Diver 2.4 software (Bioptigen). After the experiment, the animals were euthanized with CO2 gas and the eyes were harvested for further experiments.

**Hematoxylin-Eosin staining**. Eyes from NOD-SCID and AKT2 inhibitor-treated mice were fixed in 2.5% glutaraldehyde followed by formalin, transferred to graded ethanol and dehydrated followed by embeding in methyl methacrylate. Sections of 1 μm were cut and stained with hematoxylin and eosin and observed under a light microscope[74].

**Quantification of sub-retinal deposits**. The number of drusen-like sub-retinal deposits were counted in a masked fashion from hematoxylin-eosin images of 12 month old *Cryba1*^fl/fl, vehicle-treated *Cryba1* cKO and AKT2 inhibitor-treated *Cryba1* cKO mice retinae respectively. Quantification of drusen-like sub-retinal deposits were done from these images by using the ImageJ/NIH image analysis system in a masked fashion[75].

**Statistics and reproducibility**. Statistical analysis was performed with Microsoft Excel and GraphPad Prism 6 software for Windows, using one-way ANOVA. Group means were compared using Tukey's post hoc test, with significance being set at $P < 0.05$. For experiments with human samples, comparisons between control and AMD groups were performed by Mann–Whitney test with significance being set at $p < 0.05$, the data distribution was determined by the Shapiro-Wilk normality test. Center lines and edge lines in box plot indicate medians and interquartile range, respectively and whiskers indicate the most extreme data points. The analyses were performed on triplicate technical replicates. Results are presented as mean ± standard deviation (SD)[76].

**Reporting summary**. Further information on research design is available in the Nature Research Reporting Summary linked to this article.

## Data availability

All data generated or analysed during this study are included in this published article (and its Supplementary Information files). The raw uncropped western blot images are supplied in a single file as Supplementary Fig. 13. The source data underlying the graphs and charts presented in the main figures are available as Supplementary Fig. 14. The RNAseq data for the genes depicted in Supplementary Fig. 5 are uploaded in the NCBI GEO database (Accession Number: GSE136280).

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

## Acknowledgements

We thank Drs. Morton Goldberg (Wilmer Eye Institute, USA) and Shomi Bhattacharya (UCL Institute of Ophthalmology, UK) for critical reading and discussions regarding this manuscript. This study was funded by National Eye Institute: EY019037-S (DS), EY027691 (JTH), EY016151 (G.A.L.), and EY 08098 (NIH core P30 to Ophthalmology, UPMC), RPB/IRRF Catalyst Award for Innovative Research Approaches for AMD (D.S.), F. Hoffmann-La Roche, Ltd., Switzerland (D.S.), Jennifer Salvitti Davis Chair in Ophthalmology (D.S.), Robert Bond Welch Chair in Ophthalmology (J.T.H.), G. Edward and G. Britton Durell Chair in Ophthalmology (G.A.L.), Karl H Hagen Chair in Ophthalmology (J.Q.), Research to Prevent Blindness (Ophthalmology, UPMC and JHMI), NY.

## Author contributions

D.S. designed the study. S.G., A.P., T.V., A.W., I.B., S.H., P.S., N.S., M.Y., J.W., M.D. conducted the experiments. S.G., A.W., I.B., M.D., A.J., S.Z., S.S., T.B., T.W.M., J.T.H., S.W., A.G., D.S. analyzed the data. S.G., A.P., T.V., I.B., S.G.K., A, N.Y., S.X., J.Q., G.A.L., S.S., A.G. contributed to the human studies. S.G., A.W., S.H., S.Z., S.S., J.T.H., A.G., D.S. wrote the paper.

## Additional information

**Competing interests:** A.J. and D.S. are inventors in a US patent filed by F. Hoffmann-La Roche, Ltd., Basel, Switzerland on an AKT2 inhibitor for treatment of dry AMD. The remaining authors declare no competing interests.

