## [Peer Review File · Communications Biology]

Reviewers' comments:

Reviewer #3 (Remarks to the Author):

This study delineates a mechanism by which neutrophils become activated and initiate migration into a compromised retina where they can induce cellular damage that resembles attributes of early-stage AMD. The investigators used a mouse model that exhibits symptoms of AMD, in vitro culture systems, as well as AMD-patient samples to identify Type-III interferon (IFN λ) as a soluble factor that can induce the expression of LCN-2 via a STAT1-dependent mechanism, thereby activating neutrophils. Evidence gathered in this study showed that the activated neutrophils begin migration via LCN-2-dependent interaction with Dab2, which results in the increased presence of integrin β 1 on the cell surface. By identifying this mechanism, the authors were able to target a signaling protein (AKT2) that is a part of the pathway, thereby demonstrating that therapeutic agents (i.e. AKT inhibitors) may be implemented as a form of treatment to relieve neutrophil-mediated symptoms of AMD.

This study is a prime example of how utilizing various model systems can be informative in attempting to dissect molecular mechanisms, which in turn allows the researchers to specifically target a signaling pathway in order to develop therapeutic agents for use in the clinic. Specifically, the authors identify how the signaling molecule IFN λ can initiate a cascade where by neutrophils become activated and migrate into the retina, leading to AMD-related phenotypes. This study not only expands our knowledge with mechanistic details about AMD, a complex pathology with several disease-contributing factors, but it also identifies a way in which therapeutic interventions can be implemented to reduce symptoms of dry AMD, a disorder with no form of effective treatments.

The methodologies used in the study are adequate to test the hypotheses put forth by the authors. The findings of this study are novel, and the conclusions drawn from the experiments are justified. They will be of great relevance to the basic vision science community as well as clinicians who treat AMD patients.

Specific Comments:

- Although the study is quite comprehensive as it stands, it would very informative if the authors were able to generate iPSC-derived RPE cells from AMD patients to test if those cells produce more IFN λ relative to control iPSC-RPE cells.
- Figure 6 would benefit from a spider diagram accompanying the retinal sections in order to provide a quantitative measure of the photoreceptor degeneration.

Reviewer #4 (Remarks to the Author):

The investigators provide new data from studies of a mouse model of age-related macular degeneration (AMD). They provide convincing evidence of neutrophil infiltration into the retina and subretinal tissues, and increased expression of IFN α , IFN γ , IFN λ , CXCL1, CXCL9, ICAM-1 and VCAM-1, focusing on the contributions of IFN λ induced expression of LCN-2, STAT1, ROS, phagocytosis and NET formation over time in the Cryba1 $^{-/-}$ mice. Some comparison with human AMD is presented, though they speculate on the presence of CD66b $^{+}$ neutrophils in the retina without documentation.

They use a murine model (NOD-SCID) to present (n=1) an observation that lipocalin-2 (LCN-2)

deficient animal has less neutrophils following IFN λ injection than wildtype mouse.

Bone marrow neutrophils were treated in vitro with IFN λ and found to increase β 1 integrin on the surface, decrease the integrin intracellularly, increase adhesion to fibrinogen and migration through fibrinogen coated filters. They speculate that β 1 integrin is necessary for adhesion to fibrinogen without experimental confirmation. Others have shown that β 2 integrins are important for adhesion to fibrinogen, so the experimental confirmation of β 1 integrin's role is necessary here.

Figures 7B and 7C show increased surface expression of β 1 integrin on neutrophils from LCN-2 $^{-/-}$ mice. This seems to contradict the authors general description of low expression and function of β 1 integrin in these mice. Please explain the data in Figures 7B and C.

Figure 7D is ambiguous, therefore not convincingly showing what the authors claim.

Overall this study provides interesting and potentially important observations. The concerns expressed above need to be addressed in order to provide a completely convincing presentation of data.

Reviewer #3

This study delineates a mechanism by which neutrophils become activated and initiate migration into a compromised retina where they can induce cellular damage that resembles attributes of early-stage AMD. The investigators used a mouse model that exhibits symptoms of AMD, in vitro culture systems, as well as AMD-patient samples to identify Type-III interferon (IFN λ) as a soluble factor that can induce the expression of LCN-2 via a STAT1-dependent mechanism, thereby activating neutrophils. Evidence gathered in this study showed that the activated neutrophils begin migration via LCN-2-dependent interaction with Dab2, which results in the increased presence of integrin β 1 on the cell surface. By identifying this mechanism, the authors were able to target a signaling protein (AKT2) that is a part of the pathway, thereby demonstrating that therapeutic agents (i.e. AKT inhibitors) may be implemented as a form of treatment to relieve neutrophil-mediated symptoms of AMD.

This study is a prime example of how utilizing various model systems can be informative in attempting to dissect molecular mechanisms, which in turn allows the researchers to specifically target a signaling pathway in order to develop therapeutic agents for use in the clinic. Specifically, the authors identify how the signaling molecule IFN λ can initiate a cascade where by neutrophils become activated and migrate into the retina, leading to AMD-related phenotypes. This study not only expands our knowledge with mechanistic details about AMD, a complex pathology with several disease-contributing factors, but it also identifies a way in which therapeutic interventions can be implemented to reduce symptoms of dry AMD, a disorder with no form of effective treatments.

The methodologies used in the study are adequate to test the hypotheses put forth by the authors. The findings of this study are novel, and the conclusions drawn from the experiments are justified. They will be of great relevance to the basic vision science community as well as clinicians who treat AMD patients.

Specific Comments:

Although the study is quite comprehensive as it stands, it would very informative if the authors were able to generate iPSC-derived RPE cells from AMD patients to test if those cells produce more IFN λ relative to control iPSC-RPE cells

This comment/suggestion is a bit confusing, since we have shown in our original manuscript that there is an increase in IFN λ expression in RPE taken directly from human AMD patients compared to age-matched control samples (Please see Figure 2b). It is not clear to me from the comment if the reviewer wants us to look at an iPSC line with AMD risk alleles since we have already shown data from AMD patient samples. I recently attended an AMD Systems Biology meeting organized by National Eye Institute (Drs. Sheldon Miller and Kapil Bharti were organizers), and we were told that the lines NEI is making at NYSCF (five lines homozygous for the chromosome 10 risk allele and five homozygous both for chromosome 10 and the CFH risk allele) wouldn't be available until late fall this year. And, their syngeneic control lines wouldn't be available until next year. Moreover, we have no expertise generating or working with iPSC lines. I hope the data shown in Figure 2b would

satisfy the reviewer that we have indeed shown that AMD patients do produce significantly high levels of IFN λ relative to controls.

Figure 6 would benefit from a spider diagram accompanying the retinal sections in order to provide a quantitative measure of the photoreceptor degeneration.

We have now included the spider diagrams as suggested by the reviewer for the retinal sections in Figure 6 (Figure 6aix and bix) to represent the quantitative measurement of the photoreceptor degeneration.

Reviewer #4

The investigators provide new data from studies of a mouse model of age-related macular degeneration (AMD). They provide convincing evidence of neutrophil infiltration into the retina and subretinal tissues, and increased expression of IFN α , IFN γ , IFN λ , CXCL1, CXCL9, ICAM-1 and VCAM-1, focusing on the contributions of IFN λ induced expression of LCN-2, STAT1, ROS, phagocytosis and NET formation over time in the Cryba1^{-/-} mice. Some comparison with human AMD is presented, though they speculate on the presence of CD66b⁺ neutrophils in the retina without documentation.

The reviewer is correct that we failed to show CD66b⁺ neutrophils in human samples. We have now included immunofluorescence data showing CD66b⁺ cells in both normal and AMD human tissue sections. However, IL28R1⁺ cells are found only in retinal sections of AMD patients, but not in controls. In Supplementary Figure 3ci-v, we show co-localization of CD66b⁺ and IL28R1⁺ only in AMD tissue samples in the revised manuscript. Therefore, our data suggests that activated neutrophils home to the retina of early AMD patients.

They use a murine model (NOD-SCID) to present (n=1) an observation that lipocalin-2 (LCN-2) deficient animal has less neutrophils following IFN λ injection than wildtype mouse.

Spectral OCT images, hematoxylin-eosin staining of retinal samples and analysis of photoreceptor degeneration by spider diagram (as suggested by Reviewer 3 above) were all done on multiple NOD-SCID mouse samples (n=5). However, the reviewer is correct in pointing out that for neutrophil infiltration studies using the NOD-SCID mice, it was only n=1. We have chosen not to image additional whole eyes, and we hope the reviewer will understand as the whole eye imaging study required significant time and resources producing 1.2 million images, 4.5 terabytes of data, 100 hours of imaging, and substantial time for image processing/preparation. We developed this technique and to the best of our knowledge nobody has used such a novel technique to show infiltration of immune cells into the eye. We believe that the data are strong as they support our other findings by more established methods.

Bone marrow neutrophils were treated in vitro with IFN λ and found to increase B1 integrin on the surface, decrease the integrin intracellularly, increase adhesion to fibrinogen and migration through fibrinogen coated filters. They speculate that β 1 integrin is necessary for adhesion to fibrinogen without experimental confirmation.

Others have shown that $\beta 2$ integrins are important for adhesion to fibrinogen, so the experimental confirmation of $\beta 1$ integrin's role is necessary here.

In our original submission, we provided data supporting the role of integrin $\beta 1$ in wild type neutrophil adhesion and transmigration upon $IFN\lambda$ stimulation (Figure 7e, f and Supplementary Figure 11). We have shown that silencing integrin $\beta 1$ with specific shRNA in wild type neutrophils can reduce $IFN\lambda$ -induced adhesion to fibrinogen and migration across fibrinogen-coated filters, without altering cell viability, suggesting that surface expression of integrin $\beta 1$ is essential for neutrophil adhesion and transmigration (Figure 7e, f and Figure S11). These data are in line with previously published studies by others, where surface expression of integrin $\beta 1$ was shown to be essential for neutrophil adhesion, extravasation and tissue homing during inflammation (PMID: 10688841, PMID: 9625769, PMID: 22683734).

Figures 7B and 7C show increased surface expression of $\beta 1$ integrin on neutrophils from LCN-2^{-/-} mice. This seems to contradict the authors general description of low expression and function of $\beta 1$ integrin in these mice. Please explain the data in Figures 7B and C.

We agree with the reviewer that Figures 7b and 7c are indeed confusing. However, several published studies showed convincing data that various independent signaling cascades could modulate integrin $\beta 1$ surface expression in neutrophils during inflammation (PMID: 25278585, PMID: 15385257, PMID: 26554893, PMID: 19118219). We have not directly tested multiple signaling cascades that could influence neutrophil surface expression in our model system, however we speculate that the increased expression of integrin $\beta 1$ on neutrophils from LCN-2^{-/-} mice that we report here could be a trigger initiated by a signaling cascade other than via $IFN\lambda$ /LCN-2/Dab2 pathway. We have now discussed this possibility in our revised manuscript (please see Page 13, Line 278-285).

Figure 7D is ambiguous, therefore not convincingly showing what the authors claim.

We agree with the reviewer that Figure 7d is unclear and does not convincingly show internalization of surface integrin $\beta 1$ in neutrophils exposed to $IFN\lambda$. Therefore, we validated integrin $\beta 1$ internalization in $IFN\lambda$ -exposed wild type or LCN-2^{-/-} neutrophils by flow cytometry. The new data convincingly shows alterations in the intracellular levels of integrin $\beta 1$ in the $IFN\lambda$ -exposed wild type neutrophils compared to control. Moreover, there was no noticeable change in the $IFN\lambda$ treated-neutrophils from LCN-2^{-/-} mice, relative to untreated LCN-2^{-/-} neutrophils (Fig. 7d). In the revised manuscript this new data has been included replacing previous Figure 7d.

Review of COMMSIO-19-0218A: Neutrophils homing into the retina trigger pathology in early age-related macular degeneration

This study delineates a mechanism by which neutrophils become activated and initiate migration into a compromised retina where they can induce cellular damage that resembles attributes of early-stage AMD. The investigators used a mouse model that exhibits symptoms of AMD, in vitro culture systems, as well as AMD-patient samples to identify Type-III interferon (IFN λ) as a soluble factor that can induce the expression of LCN-2 via a STAT1-dependent mechanism, thereby activating neutrophils. Evidence gathered in this study showed that the activated neutrophils begin migration via LCN-2-dependent interaction with Dab2, which results in the increased presence of integrin β 1 on the cell surface. By identifying this mechanism, the authors were able to target a signaling protein (AKT2) that is a part of the pathway, thereby demonstrating that therapeutic agents (i.e. AKT inhibitors) may be implemented as a form of treatment to relieve neutrophil-mediated symptoms of AMD.

This study is a prime example of how utilizing various model systems can be informative in attempting to dissect molecular mechanisms, which in turn allows the researchers to specifically target a signaling pathway in order to develop therapeutic agents for use in the clinic. Specifically, the authors identify how the signaling molecule IFN λ can initiate a cascade where by neutrophils become activated and migrate into the retina, leading to AMD-related phenotypes. This study not only expands our knowledge with mechanistic details about AMD, a complex pathology with several disease-contributing factors, but it also identifies a way in which therapeutic interventions can be implemented to reduce symptoms of dry AMD, a disorder with no form of effective treatments.

The methodologies used in the study are adequate to test the hypotheses put forth by the authors. The findings of this study are novel, and the conclusions drawn from the experiments are justified. They will be of great relevance to the basic vision science community as well as clinicians who treat AMD patients.

Specific Comments:

- The authors have addressed all of my concerns in their revised manuscript.